# Sulfheme formation during homocysteine S-oxygenation by catalase in cancers and neurodegenerative diseases

Dominique Padovani[1], Assia Hessani[1], Francine T. Castillo[2], Géraldine Liot[3], Mireille Andriamihaja[4], Annaïg Lan[4], Camilla Pilati[5], François Blachier[4], Suvajit Sen[2], Erwan Galardon[1] & Isabelle Artaud[1]

Accumulating evidence suggests that abnormal levels of homocysteine are associated with vascular dysfunctions, cancer cell proliferation and various neurodegenerative diseases. With respect to the latter, a perturbation of transition metal homeostasis and an inhibition of catalase bioactivity have been reported. Herein, we report on some of the molecular bases for the cellular toxicity of homocysteine and demonstrate that it induces the formation of sulfcatalase, an irreversible inactive state of the enzyme, without the intervention of hydrogen sulfide. Initially, homocysteine reacts with native catalase and/or redox-active transition metal ions to generate thiyl radicals that mediate compound II formation, a temporarily inactive state of the enzyme. Then, the ferryl centre of compound II intervenes into the unprecedented S-oxygenation of homocysteine to engender the corresponding sulfenic acid species that further participates into the prosthetic heme modification through the formation of an unusual Fe(II) sulfonium. In addition, our *ex cellulo* studies performed on cancer cells, models of neurodegenerative diseases and ulcerative colitis suggest the likelihood of this scenario in a subset of cancer cells, as well as in a cellular model of Parkinson's disease. Our findings expand the repertoire of heme modifications promoted by biological compounds and point out another deleterious trait of disturbed homocysteine levels that could participate in the aetiology of these diseases.

[1] UMR 8601, LCBPT, CNRS-Université Paris Descartes, Sorbonne Paris Cité, 45 rue des Sts Pères, Paris 75006, France. [2] Department of Obstetrics and Gynecology, David Geffen School of Medicine at University of California at Los Angeles, Los Angeles, California 90095, USA. [3] Neurodegenerative Diseases Laboratory, UMR9199, CEA, CNRS, Paris-Sud University, Paris-Saclay University, MIRCen, I2BM, DRF, 18 route du Panorama, B.P. 6, Fontenay-aux-Roses 92265, France. [4] UMR 914 INRA-AgroParisTech, Nutrition Physiology and Ingestive Behavior, 16 Rue Claude Bernard, Paris 75005, France. [5] INSERM UMR-S1147, CNRS SNC 5014, Université Paris Descartes, Sorbonne Paris Cité, 45 rue des Sts Pères, Paris 75006, France. Correspondence and requests for materials should be addressed to D.P. (email: dominique.padovani@parisdescartes.fr).

Imbalances in the equilibrium of thiol-compounds and their redox-based signalling pathways are often associated with severe pathologies. For instance, impairments in the metabolism of methionine (Met) and particularly in the transsulfuration pathway are responsible for homocystinuria[1–3]. Homocystinuria is characterized by elevated homocysteine (HCys) levels (mild to severe homocystinuria: 15–500 μM HCys) and is the most commonly inherited disorder in Met metabolism as well as a risk factor associated with various pathologies such as vascular inflammation (cardiovascular disease, stroke, thrombosis) or neural tube defect[1,4–6]. In addition, disturbed HCys levels are associated with the direct or indirect perturbation of redox homeostasis, with the cell proliferation rates in various tumour cells and with diverse neurodegenerative diseases[7–11]. Notably, these latter pathologies display a perturbation of transition metal homeostasis and a deregulation of the enzymatic and chemical antioxidant systems. In particular, patients or animals bearing tumours, or animal models of chronic hyperhomocystinemia exhibit a dysfunctional catalase bioactivity[12–18].

Mammalian catalase (CAT) is a well-known homotetrameric peroxisomal Fe-protoporphyrin IX (PPIX) containing enzyme that is essential in protecting the cell from oxidative damage at high $H_2O_2$ levels[19]. The canonical activity of CAT lies within the two step conversion of hydrogen peroxide ($H_2O_2$) into dioxygen and water (equations (1) and (2)). First, CAT–Fe(III) reduces a molecule of $H_2O_2$ into water with the concomitant formation of compound I, that is, CAT–Fe(IV) = O plus a porphyrin radical cation (equation (1)). Second, compound I oxidizes a second molecule of $H_2O_2$ into water and dioxygen (equation (2)).

$$(PPIX) - CAT - Fe(III) + H_2O_2 \rightarrow$$
$$^{+\bullet}(PPIX) - CAT - Fe(IV) = O + H_2O \tag{1}$$

$$^{+\bullet}(PPIX) - CAT - Fe(IV) = O + H_2O_2 \rightarrow$$
$$(PPIX) - CAT - Fe(III) + O_2 + H_2O \tag{2}$$

CAT also reduces $H_2O_2$ to water without dioxygen production at low-physiological $H_2O_2$ concentrations[20] and it exhibits a peroxidatic activity towards low-molecular weight alcohols[21]. Moreover, CAT participates in the metabolism of endogenous substrates and carcinogens via its $H_2O_2$-independent oxidase activity[22]. Due to its pivotal role in the antioxidant defence system, the reactivity of CAT with small natural ligands has been extensively studied. The activity of CAT is impaired by small molecules through the formation of either a Low Spin (for example, hydrogen sulfide, cyanide) or a High Spin (for example, formate) iron-complex[19,23–25]. Interestingly, CAT also interacts in vitro with many natural (for example, cysteine (Cys), glutathione (GSH)) or non-natural (for example, 2-mercaptoethanol, dithiothreitol) sulfhydryl compounds and exhibits two types of reactivity in terms of changes in absorption spectra with either the formation of an uncharacterized inactive catalase type I or the generation of an inactive catalase type II (CAT–Fe(IV) = O)[26,27]. However, the underlying mechanisms that govern the reactivity of CAT towards thiol-containing compounds are still not fully understood.

An increase in the levels of the metabolic compound HCys is a common feature of various forms of cancer and several neurodegenerative diseases[7–11]. However, a divergent trend regarding the metabolic dysfunction of other sulfhydryl compounds such as hydrogen sulfide ($H_2S$) is observed in these pathologies[28,29]. Accordingly, we posit that an impairment in the levels of HCys, associated with a perturbation of transition metal homeostasis, could play a pivotal role in the aetiology of cancers and neurodegenerative diseases through the inactivation of CAT bioactivity, changes in hydrogen peroxide homeostasis and its signalling pathways.

We report here that HCys, Cys and GSH inhibit the activity of CAT in vitro and display pathologically relevant relative half inhibitory concentrations ($IC_{50}$) values only in the presence of redox-active metal ions. The reactivity of CAT with these thiol-containing compounds (RSH) takes place through two types of reaction pathways that can be followed by changes in absorption spectra. First, RSH are oxidized to thiyl radicals by native CAT–Fe(III) and/or redox-active transition metal ions. Thiyl radicals then enter a futile redox cycling that mediates compound II (CAT–Fe(IV) = O) formation, a temporarily inactive state of the enzyme. Second, compound II intervenes in an unprecedented S-oxygenation reaction only in the presence of HCys. The direct O atom transfer from compound II to the S atom of HCys results in the corresponding sulfenic acid (RSOH) species that further participates in the prosthetic heme modification through the formation of an unusual Fe(II) sulfonium. The latter displays some unique spectral properties and reactivity towards oxidants such as $O_2$ and $H_2O_2$ and it undergoes oxidation followed by C–S bond cleavage to give vinylglycine along with CAT–Fe(III) sulfheme, an irreversibly inactive state of the enzyme. At last, we performed experiments on colorectal and breast cancer cells, various cellular models of neurodegenerative diseases and colitis to examine if the formation CAT–Fe(III) sulfheme takes place under pathological conditions. Our in vitro and ex cellulo studies suggest that this scenario is most likely to occur in numerous cancers as well as in a cellular model of Parkinson's disease. Our findings support the evidence that sulfheme formation can occur without $H_2S$ intervention via the unprecedented S-oxygenation of HCys by a heme-oxo-iron(IV). In addition, the results from this study not only critically expand the scope of prosthetic heme modifications induced by biological compounds but also enlarge the adverse functions associated with disturbed HCys levels, particularly in combination with perturbed homeostasis of redox-active transition metal ions.

## Results

**Inhibition of catalase activity by sulfhydryl compounds.** As reported earlier, the activity of catalase is defective in pathological disorders presenting abnormal levels of HCys. We therefore hypothesized that high levels of this metabolic compound could play a role during the inactivation of CAT bioactivity observed in various diseases. Accordingly, we monitored the activity of CAT as a function of the concentration of thiols (RSH) in the presence (that is, physiological conditions) or absence (that is, pathological conditions) of the chelating agent diethylene triamine pentaacetic acid (DTPA). In the presence of DTPA, which does not interfere with the activity of CAT (see Supplementary Fig. 1), CAT is partially inactivated by RSH with non physiological relative $IC_{50}$ values of $0.59 \pm 0.07$ mM, $1.44 \pm 0.37$ mM and $21.2 \pm 8.5$ mM for Cys, HCys or GSH, respectively (Fig. 1a). Accordingly, CAT is not inhibited by biological thiols under physiological conditions. In addition, the partial inhibition ($\sim 50\%$) of CAT activity by RSH advocates for half-site reactivity, as reported with aminotriazole[19].

Various forms of cancer and several neurodegenerative diseases exhibit homocystinuria as well as a perturbed homeostasis of transition metal ions such as iron and copper[7–11]. Consequently, we monitored the activity of CAT as a function of RSH concentration in the presence of iron (55–145 nM) used to mimic a deregulation of transition metal homeostasis (Fig. 1b). Redox-active transition metal ions have a significant effect on the inhibitory capacity of biological thiol-containing compounds, leading to a drastic drop (30–260-fold) in their relative $IC_{50}$ values to $3.5 \pm 0.3$ μM, $37 \pm 3$ μM and $55 \pm 14$ μM for Cys, HCys

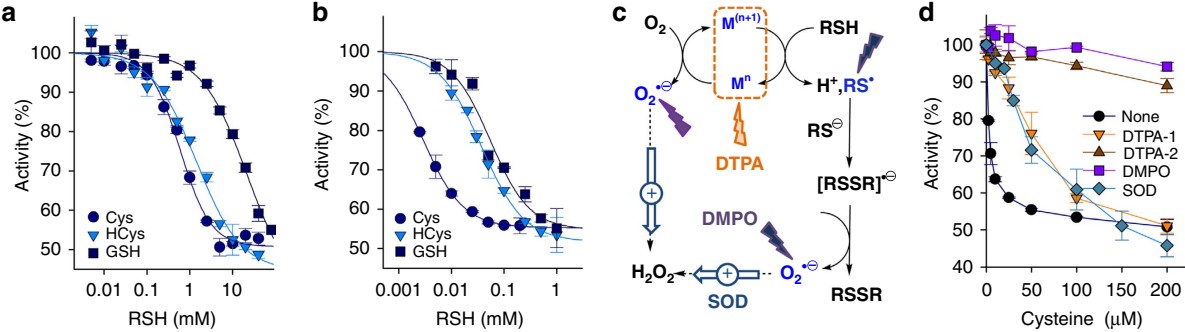

**Figure 1 | The activity of catalase is inhibited by biological thiols. (a,b)** Representative experiments showing the dependency of CAT activity on the concentration of biological thiols in the absence (**a**, +1 mM DTPA) or presence (**b**, − DTPA) of 55–145 nM iron in 50 mM phosphate buffer at pH 7.4 and 25 °C. The individual plots were fitted with a four parameter logistic equation and the relative $IC_{50}$ values ($n = 3 \pm$ s.d.) are reported in the text. (**c**) Futile redox cycle of biological thiol-containing compounds induced by redox-active transition metal ions ($M^{(n+1)}/M^n$). The effects of various additives (chelating agent diethylene triamine pentaacetic acid (DTPA), nitrone spin trapping agent 5,5-Dimethyl-1-Pyrroline-N-Oxide (DMPO) and superoxide dismutase (SOD)) are shown. (**d**) Dependency of CAT activity as a function of Cys concentration in the presence of iron (55–145 nM) and various additives in 50 mM phosphate buffer at pH 7.4 and 25 °C. The relative $IC_{50}$ values extracted from the analysis of the plots are reported in the text ($n = 3 \pm$ s.d.).

or GSH, respectively (Fig. 1b). Interestingly, these $IC_{50}$ values now fall within the range of RSH concentrations observed in physiological or pathological conditions. This strongly suggests that sulfhydryl compounds and redox-active transition metal ions may cooperate to mediate the inactivation of catalase bioactivity in a variety of diseases. In the presence of copper or iron, RSH can enter into a futile redox cycling to generate thiyl radicals (RS●), superoxide anion radicals ($O_2^{●−}$) and $H_2O_2$ which originates from the dismutation of the latter (Fig. 1c). To determine which of these species is responsible for the aforementioned severe drop in the relative $IC_{50}$ values, we monitored the activity of CAT as a function of the concentration of Cys in the presence of iron (55–145 nM) and various additives (Fig. 1d). Cys exhibits relative $IC_{50}$ values of $3.5 \pm 0.3 \mu M$, $69 \pm 27 \mu M$ or $594 \pm 67 \mu M$ in the absence or presence of 0.2 mM or 1 mM DTPA, respectively. These results confirm the preponderant role of redox-active transition metal ions in the futile redox cycling of RSH and the inhibition of CAT bioactivity under pathological conditions. In the presence of the nitrone spin-trap 5,5-Dimethyl-1-Pyrroline-N-Oxide (DMPO), Cys now displays a relative $IC_{50}$ value $> 700 \mu M$, suggesting that RS● and/or $O_2^{●−}$ mediate the inhibition of CAT activity. Finally, superoxide dismutase (SOD) enhances the relative $IC_{50}$ for Cys from $3.5 \pm 0.3 \mu M$ to $66 \pm 16 \mu M$. This strongly suggests that superoxide anion radicals also participate in the inactivation of CAT activity, in agreement with $O_2^{●−}$ contribution to the inactivation of CAT bioactivity in human breast cancer (HBC) cells[18].

**Homocysteine induces sulfheme formation.** The interaction of CAT with thiol-containing compounds can take place through two types of reactivity in terms of changes in absorption spectra[26,27]. We therefore monitored spectral changes over time when CAT was allowed to react with RSH (Fig. 2; Supplementary Figs 2 and 3). The ultraviolet–visible spectra recorded during the reactivity of native CAT–Fe(III) and HCys in the presence of DTPA clearly show the formation of several distinct species (Fig. 2a–c). Prominently, similar spectral changes take place under settings mimicking pathological conditions (see Supplementary Fig. 2). The first transient species (Soret band at 421 and α-bands at 528 and 567 nm), produced from CAT–Fe(III) with clear isosbestic points at 419, 482, 520, 604 and 650 nm, exhibits distinct spectral features characteristic of compound II CAT–Fe(IV)=O (Fig. 2a)[30]. This ferryl intermediate is transformed into a second transient species,

typical of an Fe(II) species (Soret band at 411 nm), with clear isosbestic points at 418, 486 and 580 nm. This Fe(II) species displays α-bands at 591, 636 and 658 nm and shares some spectral similarities with CAT–Fe(II) (α-bands at 561 and 595 nm) or ferrosulfcatalase (α-band at 635 nm; Fig. 2b)[24,31]. Its exposure to CO generates a new species ($\lambda_{max} = 412$ and 626 nm) with spectral properties that differ from the classical 6-coordinate CAT–Fe(II)-CO ($\lambda_{max} = 425$, 544 and 570 nm) species but are fairly analogous to the one ascribed to carboxyferrosulfcatalase (α-band at 627 nm; Fig. 2d)[24,31]. These observations suggest that the second species corresponds to an unusually stable Fe(II) sulfcatalase-like species. Finally, the second intermediate is further converted with isosbestic points at 422, 479 and 535 nm into an end product ($\lambda_{max} = 404$, 585 and 710 nm) that does not react with CO (Fig. 2e), and this species displays spectral features akin to ferric sulfcatalase (Fig. 2c)[24].

To categorically identify the final product, we performed high-performance liquid chromatography–high-resolution mass spectrometry (HPLC–HRMS) and HPLC coupled to mass spectrometry (HPLC–MS/MS) analyses of the heme-iron prosthetic group after its extraction with butan-2-one under acidic conditions (Fig. 2f; Table 1; Supplementary Fig. 4). Notably, the extracted heme-iron prosthetic group is rather unstable ($t_{1/2} = 13.2 \pm 0.3$ min at 20 °C) and the addition of imidazole immediately after its extraction stabilizes it ($t_{1/2} = 166 \pm 35$ min at 20 °C; Supplementary Fig. 5), thus permitting HPLC–HRMS and HPLC–MS/MS analyses. The high-resolution mass spectrum (ESI+) of the heme-iron extracted from CAT reacted with HCys shows an additional molecular ion at $m/z = 648.1486$ [M + H$^+$] in comparison with the one extracted from native CAT ($m/z = 616.1766$ [M + H$^+$]; Fig. 2f). The molecular mass of this extra-component matches that predicted for a sulfheme-iron derivative (calculated $m/z = 648.1494$ [M + H$^+$]; $\Delta$mmu $= -0.220$; Table 1) very well. These results clearly demonstrate that the final product is sulfcatalase and that HCys induces the production of a sulfheme species. It is worth mentioning that we were unable to characterize this sulfheme species by electron paramagnetic resonance (EPR) spectroscopy due to the half-site reactivity of CAT with RSH. Hence, the High Spin EPR signal resulting from the reactivity of CAT with HCys clearly differs from the one observed with native CAT–Fe(III) alone but appears to be a mixture of several High Spin species (see Supplementary Fig. 6).

To determine if HCys is the only biological thiol that induces the formation of sulfcatalase, we monitored the spectral changes

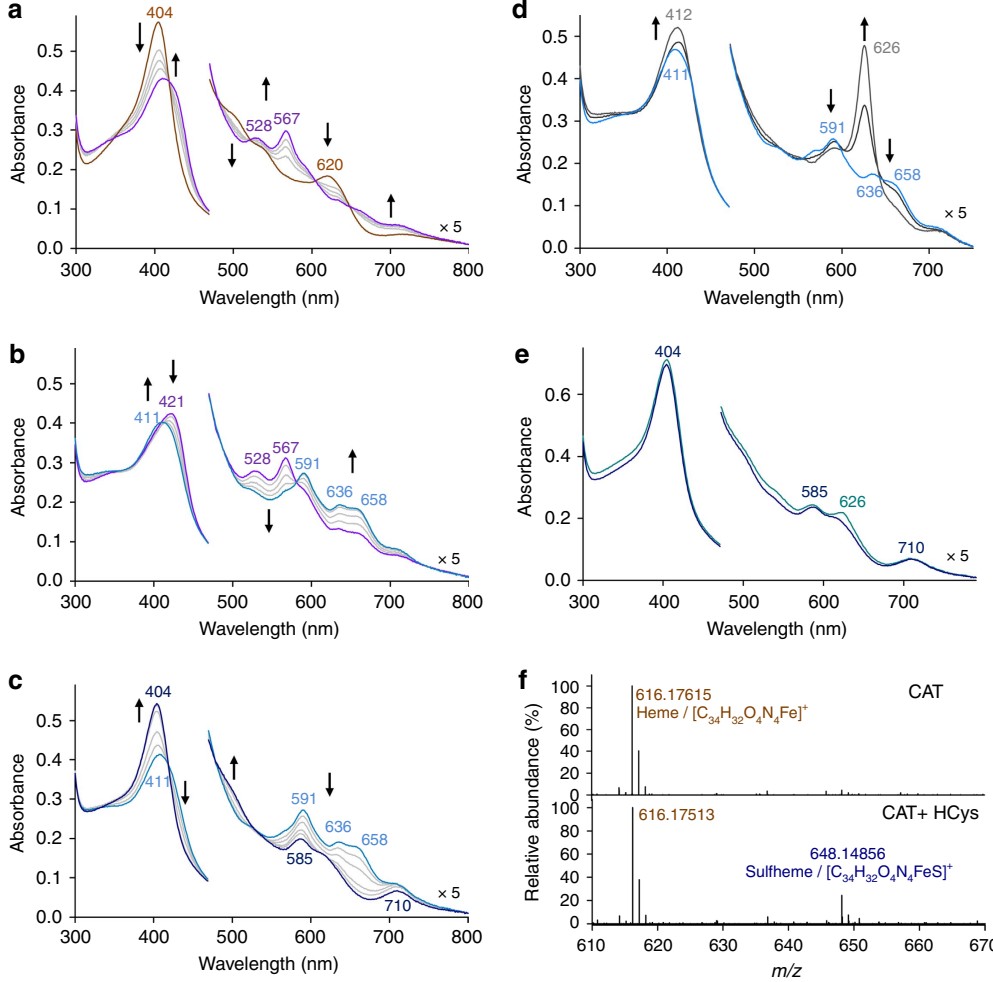

**Figure 2 | HCys induces sulfheme formation. (a–c)** Ultraviolet–visible spectral changes recorded over time (**a**, 0–30 min; **b**, 60–180 min; **c**, 210–480 min) when catalase (1.75 μM) reacts with HCys (2 mM) in 50 mM KPi, pH 7.4, 1 mM DTPA at 25 °C. (**d,e**) Spectral changes monitored by ultraviolet–visible spectroscopy when the second intermediate (**d**) or the end product (**e**) is incubated with carbon monoxide. (**f**) High-resolution mass spectra (ESI$^+$) of the heme-iron prosthetic group after its extraction from CAT (top) or CAT reacted with 2 mM HCys (bottom).

| **Table 1 | List of compounds detected by HPLC–HRMS.** | | | |
|---|---|---|---|
| **Compounds** | **Mode** | **$m/z$ theory** | **$m/z$ expt** | **Δmmu** |
| Heme iron | ESI + | 616.1773 | 616.1766 | − 0.189 |
| Sulfheme iron | ESI + | 648.1494 | 648.1486 | − 0.220 |
| HCys | ESI + | 136.0432 | 136.0422 | − 0.456 |
| Homocystine | ESI + | 269.0629 | 269.0616 | − 0.875 |
|  | ESI − | 267.0473 | 267.0479 | + 0.068 |
| Dimedone-HCys | ESI + | 274.1113 | 274.1096 | − 1.135 |
|  | ESI − | 272.0957 | 272.0961 | + 0.975 |
| Dimedone-HCys-K | ESI + | 312.0672 | 312.0654 | − 1.237 |
| Vinylglycine | ESI + | 100.0399 | 100.0393 | − 0.015 |
| 2-aminobutyric acid | ESI − | 102.0555 | 102.0549 | − 0.025 |
| α-ketobutyrate | ESI − | 101.0239 | 101.0233 | − 0.051 |
| BCN-HCys | ESI + | 286.1477 | 286.1468 | − 0.381 |
| BCN-HCys (S$^+$O$^-$) | ESI + | 302.1426 | 302.1419 | − 0.205 |

that occurred when CAT reacted with Cys or GSH in the presence or absence of DTPA (see Supplementary Fig. 3). Similar intermediates to those observed with HCys (Fig. 2a–c) accumulate with high Cys concentrations only in the presence of DTPA (see Supplementary Fig. 3a). Sulfheme formation was further confirmed by HPLC–MS/MS analysis after the extraction of the heme-iron (see Supplementary Fig. 4). Surprisingly, the reactivity

of CAT with a physiological concentration of Cys in the presence of iron only leads to the reversible formation of compound II (see Supplementary Fig. 3b,c), thus suggesting that CAT–Fe(IV) = O represents a temporarily inactive state of the enzyme. Similarly, compound II formation was only observed when CAT is incubated with GSH (see Supplementary Fig. 3d). Collectively, these results suggest that HCys is the sole biological sulfhydryl compound able to induce sulfheme formation under pathological conditions.

**RSH oxidation mediates the formation of compound II.** Next, we focused on obtaining mechanistic insights into the formation of sulfheme formation (Fig. 3). At first, we attained deeper knowledge of the generation of compound II (Fig. 3b) that corresponds to the temporarily inactive state of the enzyme, as described above. The reaction between native CAT–Fe(III) and thiol-containing compounds begins with the reduction of CAT–Fe(III) by RSH (Fig. 3b, (**1**)–(**4**)). This process first takes place through the coordination of RSH to the metal ion, generating the Low Spin iron-complex CAT–Fe(III)–RSH (**2**). The deprotonation of the bound-thiol by the distal histidine nearby the heme-iron produces an unstable CAT–Fe(III)–RS$^-$ complex (**3**) that is rapidly converted to CAT–Fe(II) (**4**) along with a thiyl radical RS•. The thiyl radicals then enter in the

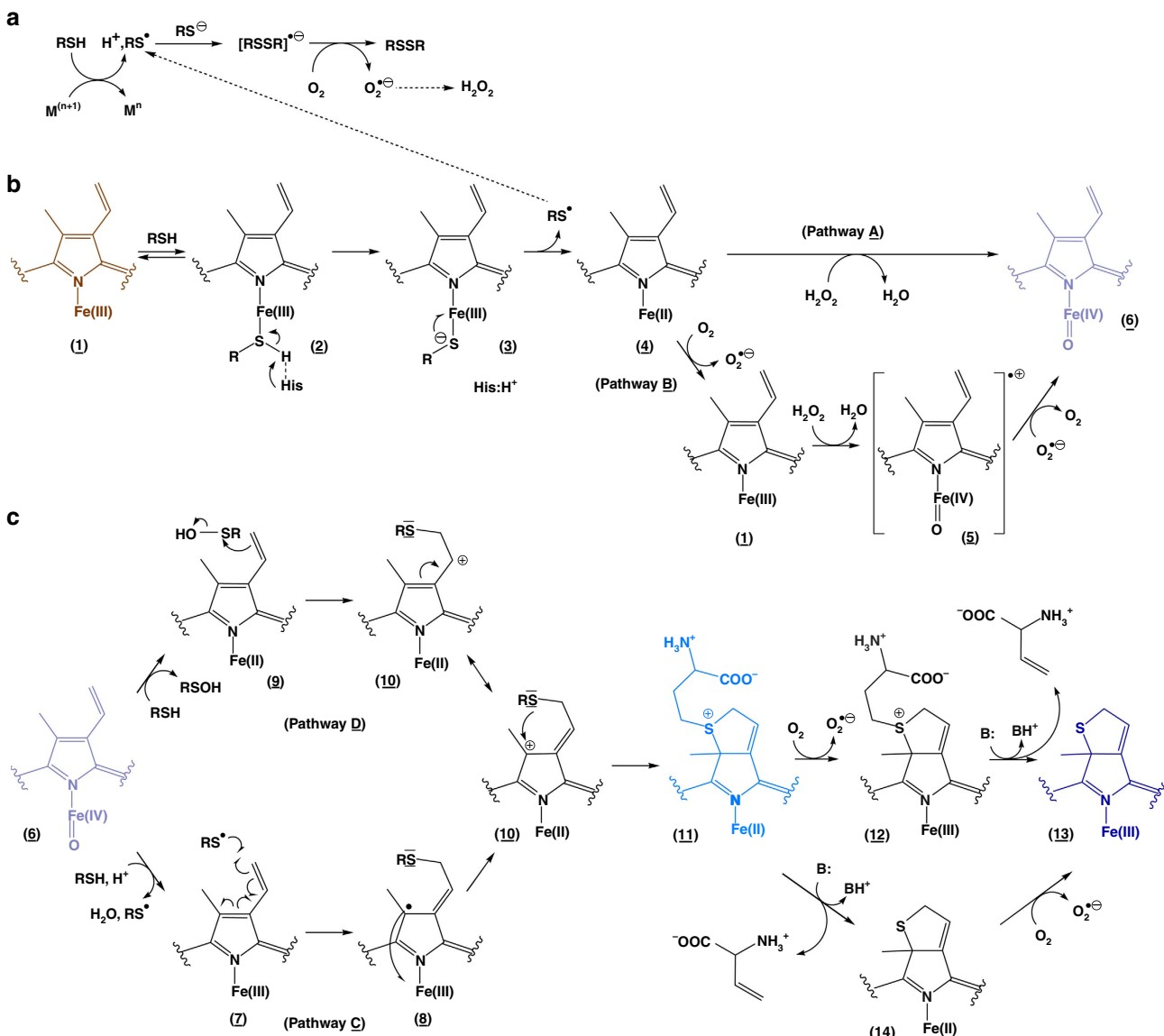

**Figure 3 | Possible reaction routes for HCys-induced sulfheme formation.** (**a**) In the presence of copper or iron, RSH can enter into a futile redox cycling to generate thiyl radicals (RS•), superoxide anion radicals (O₂• − ) and $H_2O_2$ which originates from the dismutation of the latter. (**b**) Possible mechanisms for the formation of compound II from CAT–Fe(III) in the presence of RSH. (**c**) Possible mechanisms for the generation of sulfheme from compound II in the presence of RSH. In here RSH represents HCys.

aforementioned futile redox cycle and participate in the production of O₂• − and $H_2O_2$ along with RS–SR formation (Figs 1c and 3a). At this point, the formation of compound II can occur via at least two different plausible scenarios (Fig. 3b, pathways A–B). In one pathway, CAT–Fe(III) (**1**) resulting from the oxidation of CAT–Fe(II) (**4**) by molecular oxygen can react with $H_2O_2$ to generate compound I (**5**). Compound I is subsequently reduced to compound II (**6**) by O₂• − ($k \sim 5 \times 10^6\,M^{-1}\,s^{-1}$, $E'^0(O_2/O_2•−) = −0.16\,V$ at pH 7; pathway B)[32,33]. Alternatively, compound II may be produced from the reaction between CAT–Fe(II) (**4**) and $H_2O_2$ (pathway A)[34]. However, this pathway seems unlikely as superoxide anion radicals directly participate in the inactivation of CAT activity (Fig. 1d). To confirm that compound II generation happens via pathway B, we carried out analytical studies, comparative kinetic studies in the presence of various additives, and we correlated these kinetics studies to the time course for catalase inactivation (Fig. 4; Tables 1 and 2; Supplementary Fig. 7).

Neither the Low Spin complex CAT–Fe(III)–HCys (**2**) nor CAT–Fe(II) (**4**) are observed by ultraviolet–visible spectroscopy during the time course of compound II formation (Fig. 2a), and so we focused on demonstrating the production of HCys• during the generation of CAT–Fe(IV) = O. Accordingly, we performed similar experiments to those described in Fig. 2 in the presence of the bioconjugation reagent bicyclo[6.1.0]nonyne (BCN) and we subsequently analysed the reaction mixture by HPLC–HRMS (Fig. 4a) and HPLC–MS/MS (see Supplementary Fig. 7b). Our HPLC–HRMS analysis reveals the presence of a molecular ion at $m/z = 286.1468$ [M + H⁺] (Fig. 4a) that corresponds well to a BCN-HCys adduct (Table 1). This adduct, though not detected in the absence of CAT, results from the addition of HCys• to BCN according to the thiol-yne reaction[35]. Its presence in the reaction mixture therefore clearly confirms the reduction of CAT–Fe(III) by RSH into CAT–Fe(II) and HCys• (Fig. 3b, (**1**) to (**4**)).

To further confirm the presence of HCys• when native CAT–Fe(III) reacts with HCys, we also performed EPR spin-trap

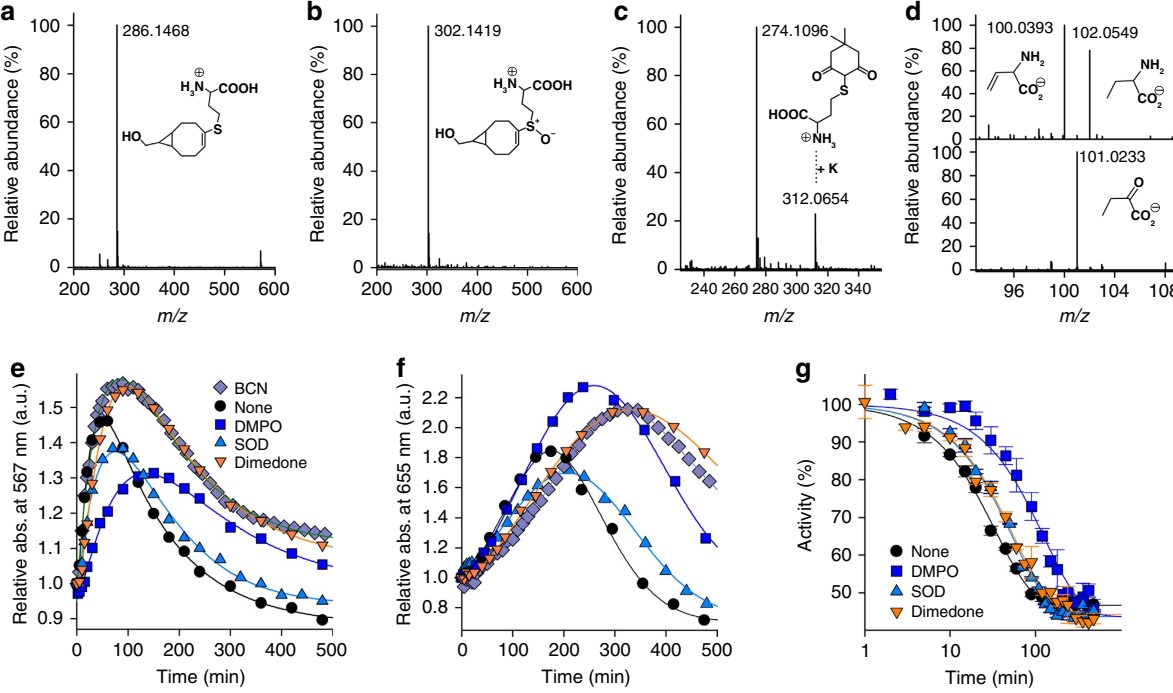

**Figure 4 | Thiyl radicals and sulfenic acid species intervene during sulfheme formation.** (**a–d**) High-resolution mass spectra of the BCN (**a,b**) and dimedone (C) adducts (ESI$^+$) as well as the HCys derivatives (D; ESI$^-$) formed during the reactivity of catalase (3 μM) with HCys (2 mM). (**e,f**) Comparative kinetic studies of CAT–Fe(IV)=O formation (**e**) and CAT–Fe(II) sulfonium generation (**f**) when catalase (1.75 μM) is exposed to 2 mM HCys in the absence or the presence of various additives. (**g**) Time course of catalase inactivation observed under the same experimental conditions as those described in (**e,f**). The $k_{inact}$ values extracted from the analysis of the plots are reported in Table 2 (n = 2 ± s.d.).

**Table 2 | Summary of catalase inactivation parameters in the presence or absence of various additives*.**

| Additives | Activity test | Ultraviolet–visible spectroscopy | |
|---|---|---|---|
| | $k_{inact}$, min$^{-1}$ | $k_{obs}(6)$, min$^{-1}$ | $k_{obs}(11) \times 10^3$, min$^{-1}$ |
| None | 0.029 ± 0.002 | 0.026 ± 0.003 | 7.88 ± 0.62 |
| DMPO | 0.009 ± 0.001 | 0.010 ± 0.002 | 5.33 ± 0.29 |
| SOD | 0.018 ± 0.001 | 0.020 ± 0.002 | 6.73 ± 0.39 |
| Dimedone | †0.018 ± 0.002 | †0.019 ± 0.003 | 4.17 ± 0.23 |
| BCN | ND | 0.018 ± 0.002 | 4.39 ± 0.11 |

*The ultraviolet–visible spectroscopy experiments (n ≥ 2 ± s.d.) and the activity tests (n = 2 ± s.d.) were performed as described in 'Methods' section.
†Effect due to the reactivity of dimedone with $H_2O_2$ and $O_2$• − (see Supplementary Fig. 12).
ND, not determined.

experiments with the nitrone spin trapping agent DMPO. Unfortunately, we were unable to characterize the adduct DMPO–HCys by RT EPR spectroscopy. This is most likely due to the slow kinetics of HCys oxidation, the reactivity of the DMPO adducts with $H_2O_2$ present in the reaction mixture, the fast decomposition of the DMPO–HCys adduct back to HCys• and the nitrone and/or the adventitious formation of DMPO–SR adducts in accordance with the Forrester-Hepburn mechanism[36]. Regardless, we analysed the HCys derivative formed during compound II formation by HPLC–HRMS. Our analysis shows the presence of a molecular ion at $m/z = 269.0616$ [M + H$^+$] that corresponds to the homocystine disulfide RS–SR (Table 1). This strongly substantiates $O_2$• − production from RS• via the transient formation of [RSSR]• − and its subsequent reactivity with $O_2$ (Fig. 3a).

Next, we performed comparative kinetic studies of compound II formation in the absence or presence of various additives and correlated these kinetic studies to the time course of catalase inactivation (Fig. 4e–g; Table 2). The results demonstrate that inhibition of CAT bioactivity is correlated to compound II

formation (Table 2), suggesting that both processes are closely intertwined. In addition, the kinetics of CAT–Fe(IV)=O formation and the time course of catalase inactivation are both equally affected by the nitrone spin-trap DMPO (Fig. 4e,g; Table 2), thus confirming that HCys• contributes to CAT–Fe(IV)=O production and enzyme inhibition. Last, the presence of SOD into the reaction mixture significantly reduces the time course of catalase inactivation and the rate of compound II formation (Fig. 4e,g; Table 2), strongly suggesting that compound II formation occurs through the pathway *B* described in Fig. 3b. Clearly, SOD-catalysed $O_2$• − dismutation should accelerate compound II creation through pathway *A*, which is in disagreement with our observations.

**Sulfheme formation occurs without $H_2S$ intervention.** Next, we investigated the underlying mechanism for HCys-induced sulfheme formation from CAT–Fe(IV)=O. The formation of a sulfheme species is documented to occur via a yet unclear mechanism through the reaction of compound II with $H_2S$

(refs 24,37). The disproportionation of two thiyl radicals RS•
results in the formation of a thiol RSH and a thione derivative
R = S whose hydrolysis can be linked to $H_2S$ production along
with a ketone R = O (ref. 38). Furthermore, $H_2S$ may be present
as a contaminant in the solution of HCys we used during our
experiments. We therefore monitored the possible presence of
$H_2S$ with an amperometric $H_2S$ microsensor. The formation of
$H_2S$ was not detected during CAT incubation with HCys in the
presence or absence of DTPA (see Supplementary Fig. 8),
suggesting that HCys induces sulfheme formation without $H_2S$
intervention through an atypical alternative mechanism.

**S-oxygenation of HCys mediates sulfheme formation.** An
alternative mechanism for sulfheme formation without $H_2S$
participation from compound II is described in Fig. 3c,
pathway C. This mechanism initially involves the one electron
oxidation of HCys by CAT–Fe(IV) = O (**6**) to form CAT–Fe(III)
along with HCys• and the release of $H_2O$ (**7**). Notably, the
oxidation of thiols or sulfide by compound II has significant
precedent in the literature. For instance, 2-mercapto-1-methyli-
midazole is oxidized by lactoperoxidase (LPO) compound II to
the corresponding thiyl radical that further inactivates LPO-
Fe(III) by modifying the heme prosthetic group[39]. Also, the
heme-oxo-iron(IV) of myoglobin is reduced by several thiol-
compounds (Cys, GSH or N-acetylcysteine) to Fe(III)-myoglobin
along with the concomitant formation of the respective thiyl
radicals[40]. Once formed, HCys• adds to the vinyl position
to generate a modified CAT–Fe(III) species bearing a
protoporphyrinic radical (**8**). This then reduces the ferric ion to
form a cationic species (**10**) that ensures ring closure by favoring
the nucleophilic attack of the sulfur atom on the electrophilic
carbon. This step leads to the generation of a Fe(II) sulfonium
(**11**) species that can either undergo an oxidation process
followed by a β-deprotonation and an elimination reaction of
the group bonded to the sulfur atom to give Fe(III) sulfheme
((**12**) and (**13**)), or the reverse sequence, that is, C–S bond
cleavage followed by the oxidation of Fe(II) sulfcatalase (**14**) into
Fe(III) sulfcatalase (**13**).

A second alternative mechanism is described in Fig. 3c,
pathway D. The first step involves the S-oxygenation of HCys by
compound II (**6**), generating CAT–Fe(II) along with a reactive
sulfenic acid species RSOH (**9**). This mechanism is unprecedented
as the S-oxygenation of organosulfur compounds such as
alkylaryl sulfides is only mediated by compound I, for example,
from chloroperoxidase or prostaglandine H synthase[41,42]. The
S-oxidation of sulfides by direct O atom transfer from compound
I to the S atom is enantioselective and is mediated by an oxene
process in which compound I experiences a two-electron
reduction to give the native enzyme. In our case, once the
S-oxygenation of HCys by compound II has taken place,
nucleophilic attack from the peripheral vinyl position on the
electrophilic sulfur of RSOH (**9**) occurs to generate the
aforementioned cationic species (**10**). Production of the Fe(II)
sulfonium (**11**) and the Fe(III) sulfheme (**13**) species then takes
place as described above.

We first carried out comparative kinetic studies in the presence
or absence of various additives to discriminate between both
mechanisms for the creation of the Fe(II) sulfonium species. To
do this, we utilized the nitrone spin-trap DMPO for trapping
HCys• resulting from compound II reduction by HCys (Fig. 3c,
pathway C), the bioconjugation reagent BCN that is capable of
capturing thiyl radicals (Fig. 3c, pathway C) as well as sulfenic
acid species RSOH (Fig. 3c, pathway D) since BCN intervenes in
the thiol-yne reaction[35] and reacts via a concerted mechanism
with RSOH[43], and the sulfenic acid trap 5,5-dimethyl-1,

3-cyclohexanedione (dimedone). Our results clearly show that
dimedone and BCN have the most significant effect on
production of the Fe(II) sulfonium species (Fig. 4f; Table 2).
This strongly suggests that the first step leading to CAT–Fe(II)-
sulfonium (**11**) from CAT–Fe(IV) = O (**6**) consists in the
S-oxygenation of HCys by compound II to generate
CAT–Fe(II) along with a reactive sulfenic acid species
RSOH (**9**), in agreement with Fig. 3c, pathway D.

To further validate these observations, we next performed
HPLC–HRMS and HPLC–MS/MS analyses ($ESI^+$ and/or $ESI^-$
modes) of the various reaction mixtures obtained after CAT
reaction with HCys in the presence of DTPA, and in the presence
or absence of various additives. Our analytical studies confirm the
presence of the previously observed RS• (adduct BCN-HCys) and
remarkably establish the presence of RSOH (adducts dimedone-
HCys and BCN-HCys-$S^+O^-$; Fig. 4a–c; Table 1; Supplementary
Fig. 7), thus clearly demonstrating that compound II takes part in
the unprecedented S-oxygenation reaction of HCys. In addition,
vinylglycine and its byproducts[44] are detected by HPLC–HRMS
(Fig. 4d; Table 1), corroborating that a β-deprotonation and
an elimination reaction of the group bonded to the sulfur atom
occur during sulfheme formation (Fig. 3c, steps (**11**) to (**14**) or
(**12**) to (**13**)).

Next, we compared the stability of CAT bearing a Fe(II)
sulfonium (**11**) or a Fe(II) sulfheme (**14**) species to establish the
sequence leading from the Fe(II) sulfonium (**11**) to the Fe(III)
sulfheme (**13**) species, that is, oxidation followed by C–S bond
cleavage or the inverse (Fig. 3c). The classical Fe(II) sulfheme (**14**) is
obtained from the reactivity of compound II with $H_2S$ (refs 24,37)
through an unknown mechanism (see Supplementary Fig. 9). We
therefore monitored its formation by ultraviolet–visible spectro-
scopy under pathological conditions. The addition of increasing
concentrations of NaSH to CAT-containing Fe(IV) = O initially
results in the formation of Fe(II)-sulfcatalase that is further
transformed into a mixture of native CAT–Fe(III) and Fe(III)-
sulfcatalase(-sulfide)[24] in the presence of excess NaSH (Fig. 5a).
Then, we assayed these two species for activity measurement in the
presence of $H_2O_2$. While Fe(II)-sulfcatalase exhibits a gain in its
activity in comparison with the activity of CAT-containing
Fe(IV) = O, the mixture of native CAT–Fe(III) and Fe(III)-

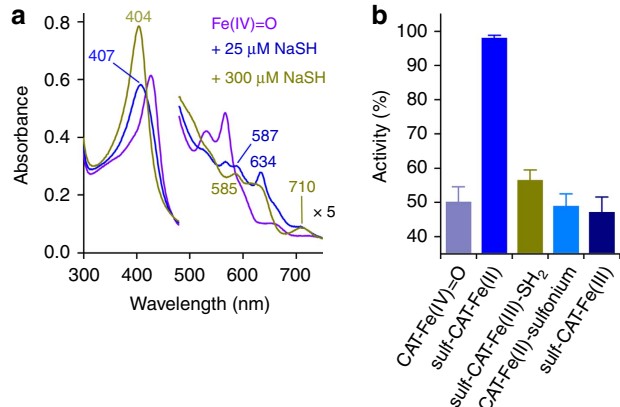

**Figure 5 | Activity of catalase containing various heme-iron species.**
(**a**) Representative experiment showing the ultraviolet–visible spectral
changes monitored at 25 °C when compound II-containing catalase
(2.4 μM) obtained under pathological conditions with HCys (200 μM)
reacts with NaSH in 50 mM KPi at pH 7.4. (**b**) Activity of the various
species obtained in (**a**) ($n = 3 ±$ s.d.). Notably, sulf-CAT-Fe(III)-SH₂
contains a mixture of native CAT–Fe(III) and sulf-CAT-Fe(III)-sulfide.
The activity of CAT-Fe(II)-sulfonium and sulf-CAT-Fe(III) were taken from
Fig. 4g.

sulfcatalase(-sulfide) displays a perturbed activity (Fig. 5b), suggesting that the Fe(III)-sulfheme(-sulfide) species is an inhibitory form of the enzyme. The observed increase in Fe(II)-sulfcatalase activity is explained by the reactivity of Fe(II) sulfcatalase with oxidants[24]. Hence, the latter species is unstable towards $O_2$ or $H_2O_2$ and regains its extra-aromaticity to produce native CAT–Fe(III) in the presence of oxidants (see Supplementary Fig. 9). In contrast, the CAT containing a Fe(II)-sulfonium species or a Fe(III)-sulfheme species derived from the former exhibits a loss in activity (Fig. 5b). These results suggest that the Fe(II) sulfonium species is insensitive to $H_2O_2$, as corroborated by ultraviolet–visible spectroscopy (see Supplementary Fig. 10), and also strongly suggest that CAT containing a Fe(II)-sulfonium species or CAT harbouring a Fe(III)-sulfheme species represent irreversible inactive states of the enzyme. In addition, these observations raise some questions about the formation of Fe(II) sulfcatalase through $H_2S$ intervention in diseases presenting with increased levels of $H_2O_2$. Finally, they strongly favour the sequence in which the Fe(II) sulfonium species (**11**) first undergoes a slow oxidation process into a Fe(III) sulfonium species (**12**) followed by a β-deprotonation and a rapid elimination reaction of the group attached to the sulfur atom to give the Fe(III) sulfheme species (**13**) and vinylglycine.

Collectively, the results from these studies substantiate the mechanism best described by Fig. 3c, pathway *D* followed by the oxidation of the Fe(II) sulfonium species and the C–S bond cleavage to give vinylglycine along with CAT–Fe(III) sulfheme (steps (**11**)–(**13**)). Thus, compound II from CAT (**6**) mediates the S-oxygenation of HCys by direct O atom transfer to the S atom of the thiol-compound to generate CAT–Fe(II) (**9**) along with a sulfenic acid derivative that further participates in electrophilic addition to the heme vinyl position (**10**). This is in full agreement with seminal work performed to apprehend the modification of the prosthetic heme from peroxidases by small molecules[45–48] and with studies conducted to analyse the modification of Fe(III) porphyrin models[49] where the oxidation of small molecules into radical species R• (NO₂•, carboxylic, azido or alkylhydrazine radicals) results in their addition to the δ-meso-position of the porphyrin moiety. In contrast, the oxidation of thiocyanate and halides ($X^-$) to electrophilic species (hypothiocyanous acid and XOH, respectively) results in their addition to the vinyl positions. Following the modification of the heme prosthetic group at the vinyl position, the cyclization at the ring periphery takes place through the formation of a Fe(II) sulfonium species (**11**) whose spectral signature is undoubtedly the α-band at 658 nm (Fig. 2b). This species is unusually stable towards oxidants ($O_2$ and $H_2O_2$), by analogy to the sulfonium ion linkage present in myeloperoxidase that stabilizes its Fe(II) form[50]. After the oxidation of the Fe(II) sulfonium species (**11**) to its Fe(III) form (**12**), the modulation of the acidity of the β protons by the sulfonium favors a base-induced deprotonation of the group bonded to the sulfur atom, thus promoting the generation of the Fe(III) sulfheme species (**13**) along with vinylglycine via an elimination reaction. This scenario parallels the involvement of sulfonium species as intermediates during the synthesis of thiophene derivatives by the cyclization of functionalized alkynes[51], during the conversion of Cys or GSH into their respective α,β-unsaturated dehydroalanyl derivatives[52–54] or during the reactivity of dibromobimane with RSH to produce a bimane thioether[55].

**Biological implications.** Thus far, the formation of sulfheme is physiologically associated with sulfhemoglobinemia, a rare condition provoking oxygen desaturation and cyanosis and resulting from the long term exposure to $H_2S$ or sulfur-containing drug

overdose[56,57]. Also, the formation of sulfheme presumably occurs during the degradation of $H_2S$ in red blood cells[58] and during the reactivity of red meat pigments with Cys-derived sulfhydryl radicals (S•⁻) under the acidic conditions of the stomach[59]. Our findings that HCys can promote sulfheme production without $H_2S$ intervention under pathological conditions raise the question of its biological relevance.

In HBC cells, the inhibition of catalase bioactivity is coupled to an increase of intracellular $H_2O_2$ levels necessary for the proliferation of cancer cells[18]. The impaired bioactivity of catalase can be partially restored on treatment with $O_2$•⁻ scavengers[18], somewhat suggesting that the inhibition of CAT activity proceeds at least through the formation of compound II (Fig. 3b, pathway *B*). However, HBC cells display increased levels of $H_2S$ that are associated with the protection of cancer cells against activated macrophages[60] and may promote the proliferation and migration of cancer cells[28]. But HBC cells also exhibit disturbed levels of HCys associated with the progression of cancer by epigenetic modulation[11]. These last observations, the significant difference in the levels of both metabolites and the previously observed stability of the Fe(II) sulfonium species *versus* the instability of the Fe(II) sulfheme species under oxidative conditions (Fig. 5), open up new possibilities for the inhibition of catalase bioactivity in cancers through HCys-induced sulfheme generation. A similar scenario may take place in ulcerative colitis and various neurodegenerative diseases, such as abnormal levels of HCys, a deregulation of the homeostasis of redox-active transition metal ions and an impairment in the production of $H_2S$ are trademarks of Alzheimer's, Parkinson's or Huntington's diseases[29,61]. Furthermore, rodent models of hyperhomo-cystinemia exhibit an increased susceptibility to colitis as well as an impaired colonic $H_2S$ synthesis[62] while the ongoing production of $H_2O_2$ contributes to epithelial dysfunction[63].

To gain insights into the likelihood of HCys-induced sulfcatalase formation in various disorders, we focused on establishing a link between this new finding and various models of pathologies. To do so, we first explored *ex cellulo* the relative specific activity of CAT (SpCAT) in various HBC cells, numerous colorectal cancer cells, diverse cellular models of neurodegenerative disorders (Hek 293T cells transfected with Htt-N171-82Q (Htt) or Alpha-synuclein-A53T (α-Syn; see Supplementary Fig. 11), and M17 human neuroblastoma cells treated with rotenone) and a rodent model of Dextran Sulfate Sodium(DSS)-induced colitis (Fig. 6a–d). SpCAT is decreased in HBC cells (Fig. 6a) and colorectal cancer cells (Fig. 6b) in comparison with control cell lines. This is in agreement with the strategy developed by cancer cells for maintaining high steady-state levels of $H_2O_2$ necessary for cell proliferation[18]. In contrast, SpCAT is activated in a cellular model of Huntington's disease (Htt) compared with the control cell line (Fig. 6c). As a substantial portion of Huntington's disease neurotoxicity results from a deficiency in the transsulfuration pathway through the interaction of Huntingtin with cystathionine β-synthase[64] and the inhibition of cystathionine-γ lyase transcription activator by mutant Huntingtin[61], the activation of SpCAT observed in our cellular model of Huntington's disease, probably due to the post-translational modification of CAT promoted by non-receptor protein tyrosine kinases[65], may participate in compensatory mechanisms activated by imbalances in the equilibrium of biological thiols. Surprisingly, SpCAT is also activated in α-Syn-expressing cells (Fig. 6c), which appears to contradict a recent study that tested the possible link between catalase inactivation and oxidative injury in brains of A53T α-Syn mice and α-Syn-expressing cells[66]. However, the activity of CAT was assessed with the means of the horseradish peroxidase fluorogenic substrate Amplex red, a method originally conceived to monitor

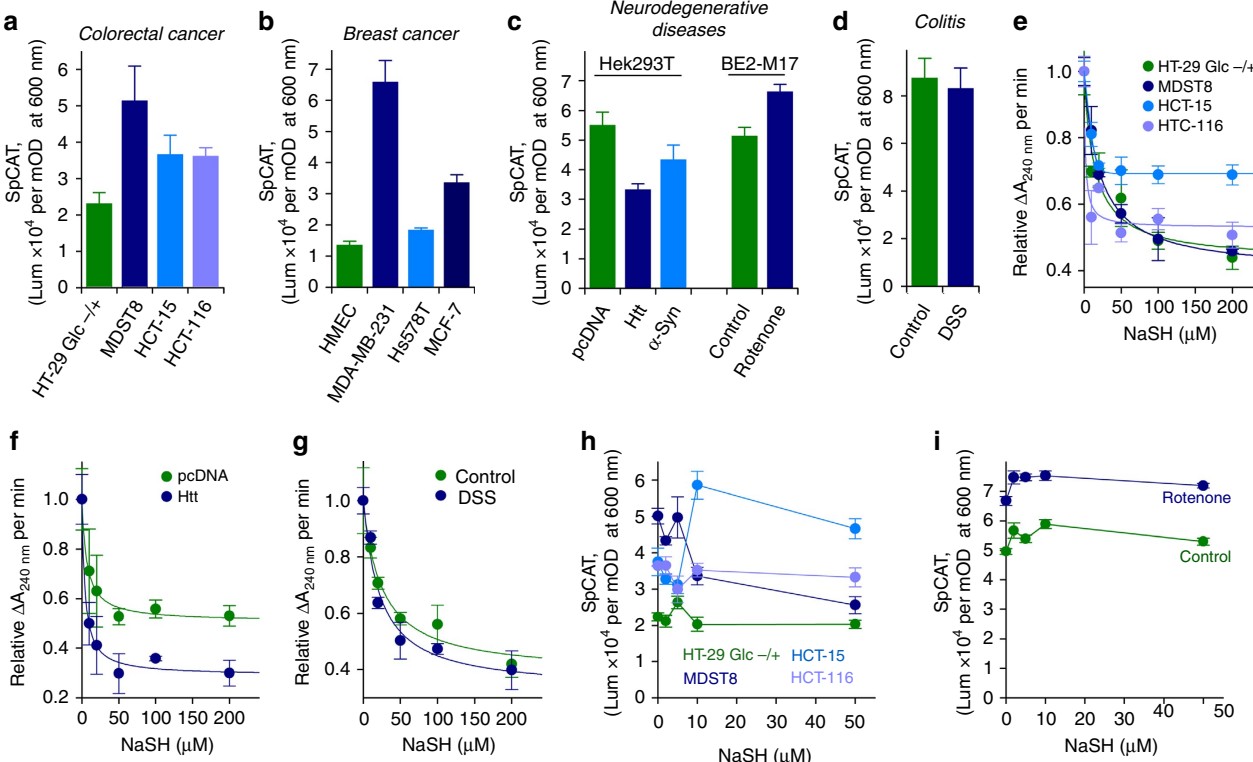

**Figure 6 | HCys-induced sulfcatalase formation is likely in various pathological disorders. (a)** Relative specific activity of immunocaptured catalase (SpCAT) in various human colorectal cancer cells. The HT-29 Glc −/+ cell line was chosen as a control as it maintains metabolic characteristics of normal colonocytes. **(b)** Comparison of SpCAT in human mammary epithelial cells (HMEC-control) and in various HBC cells. **(c)** SpCAT measured in diverse cellular models of neurodegenerative disorders: Hek 293T cells transfected with an empty plasmid (pcDNA-control), Htt-N171-82Q (Htt) or Alpha-synuclein-A53T ($\alpha$-Syn) and M17 human neuroblastoma cells treated or not (control) with 100 nM rotenone. **(d)** SpCAT in crypt and surface epithelial cells isolated from a rodent model of DSS-induced colitis (DSS) in comparison with untreated mice (control). **(e–g)** Dependency of $H_2O_2$ consumption in various cell lysates on the concentration of NaSH. Cell lysates were preincubated with NaSH on ice for 5–10 min before monitoring $H_2O_2$ disappearance at 240 nm. **(h,i)** Dependency of SpCAT on the concentration of NaSH in various human colorectal cancer cells **(h)** and in a cellular model of Parkinson's disease **(i)**. Data are represented as means ± s.d. (colorectal cancer cells and neurodegenerative diseases, $n=2$; breast cancer cells and colitis, $n=3$). Each experiment was performed in duplicate (SpCAT) or triplicate ($H_2O_2$ consumption).

reactive oxidative species that also measures the far more prevalent reactive sulfur species generated endogenously under oxidative stress conditions[67]. On the other hand, SpCAT is inhibited in our second cellular model of Parkinson's disease, that is, BE2-M17 cells treated with rotenone (Fig. 6c). In this case, the inactivation of CAT bioactivity may be related to the sensitization of dopaminergic neurons to dysfunction and death when exposed to rotenone or iron(II) in the presence of elevated HCys levels[68]. Finally, SpCAT is not modified in crypt and surface epithelial cells isolated from a rodent model of DSS-induced colitis in comparison with untreated mice (Fig. 6d), implying that the steady generation of $H_2O_2$ observed in colitis is essentially caused by the activity of dual oxidases[63].

As reported earlier, CAT–Fe(IV)=O represents a temporarily inactive state of the enzyme. It can be transformed into an active state of the enzyme via its reactivity with $H_2S$, the transient formation of a Fe(II)-sulfheme species and the loss of aromaticity of the latter to produce native CAT in the presence of oxidants (Fig. 5; Supplementary Fig. 9). In contrast, CAT-containing a Fe(III) sulfheme generated through S-oxygenation of HCys corresponds to an irreversibly inactive state of the enzyme (Fig. 5b). Accordingly, we first monitored the effect of various concentrations of NaSH on the rate of $H_2O_2$ consumption in cell lysates in an attempt to determine which heme-iron species is responsible for SpCAT inhibition in several of our cellular models (Fig. 6e–g). NaSH inhibits the rate of $H_2O_2$ consumption in all colorectal cancer cells, even in HT-29 Glc −/+ cells that maintain

metabolic characteristics of normal colonocytes, and exhibits low relative $IC_{50}$ values of $20.4 \pm 4.0 \mu M$ (MDST8 cells), $5.8 \pm 1.8 \mu M$ (HCT-15 cells), $1.8 \pm 1.5 \mu M$ (HCT-116 cells) and $15.0 \pm 5.3 \mu M$ (HT-29 Glc −/+ cells; Fig. 6e). Similarly, NaSH reduces the rate of $H_2O_2$ disappearance in pathological models that exhibit either an activated or unaffected SpCAT (Fig. 6f,g). Hence, NaSH displays relative $IC_{50}$ values of $6.5 \pm 3.6 \mu M$ or $4.0 \pm 1.5 \mu M$ for Hek 293T cells transfected with pcDNA or Htt, respectively (Fig. 6f), and relative $IC_{50}$ values of $25.2 \pm 6.7 \mu M$ or $21.6 \pm 6.2 \mu M$ for crypt and surface epithelial cells isolated from untreated and DSS-treated mice, respectively (Fig. 6g). These results suggest that $H_2S$ inhibits ubiquitously cellular antioxidant enzymes that rapidly consume $H_2O_2$, such as peroxiredoxins and GSH peroxidases, most likely via the formation of Cys persulfides that affect the activity of target proteins[69]. Regardless, these observations do not provide any information regarding the heme-iron species accountable for SpCAT inhibition in cancer cells and rotenone-treated BE2-M17 cells.

As a result, we explored the influence of NaSH on SpCAT in colorectal cancer cells and M17 human neuroblastoma cells treated with rotenone, which should allow discriminating between a temporarily (CAT–Fe(IV)=O) and an irreversibly (CAT–Fe(III)–sulfheme) inactive state of the enzyme (Fig. 6h,i). SpCAT is almost unaffected by NaSH in the control cell line HT-29 Glc −/+ as well as in HCT-116 cancer cells (Fig. 6h), and the response of SpCAT towards NaSH is similar in BE2-M17 cells and in rotenone-treated BE2-M17 cells (Fig. 6i). These results

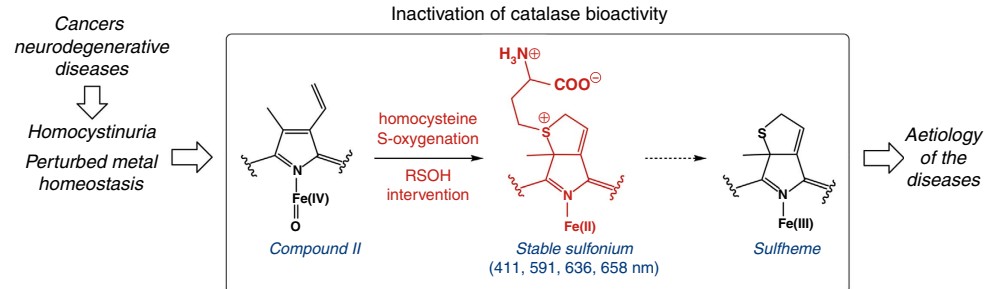

**Figure 7 | Pathological implications of elevated HCys levels.** In some forms of cancer and some neurodegenerative diseases, disturbed HCys levels, in combination with perturbed homeostasis of redox-active transition metal ions, can lead to sulfheme formation without $H_2S$ intervention. Sulfheme formation occurs through the unprecedented S-oxygenation of HCys by the heme-oxo-iron(IV) of catalase. The resulting prosthetic heme modification induces an irreversible inhibition of catalase bioactivity and may participate to the aetiology of various disorders.

suggest that the inhibition of CAT in HCT-116 cells and rotenone-treated BE2-M17 cells may occur through sulfheme formation promoted by HCys. In contrast, SpCAT is activated by NaSH in MDST8 cancer cells (Fig. 6h), indicating that the inhibition of CAT in this cell line may solely occur through the generation of a temporarily inactive state of the enzyme, that is, CAT–Fe(IV) = O. Finally, SpCAT is inhibited in HCT-15 cancer cells (Fig. 6h), most probably through the formation of an inhibitory form of the enzyme such as CAT–Fe(III)-sulfheme-sulfide (Fig. 5b).

Collectively, the results from these *ex cellulo* studies suggest that $H_2S$ inhibits ubiquitously cellular antioxidant enzymes other than catalase. In addition, the investigation of the influence of $H_2S$ on the relative specific activity of catalase allows discriminating between a temporarily and an irreversibly inactive state of the enzyme in models of pathological disorders. Finally, our results substantiate the relevance of HCys-induced sulfheme formation without $H_2S$ intervention in a subset of cancer cells as well as in a cellular model of Parkinson's disease (Fig. 7).

Overall, we investigated here the reactivity of catalase toward biological sulfhydryl compounds and in particular, HCys. HCys is a branched-point intermediate of the transsulfuration and Met salvage pathways that accumulates in pathologies presenting with homocystinuria. The key finding of this study is the unprecedented intervention of an heme-iron(IV)-oxo species into an S-oxygenation reaction, as established by ultraviolet–visible spectroscopy, activity tests, mass spectrometry and comparative kinetic studies. The direct O atom transfer from compound II to the S atom of HCys leads to the production of a sulfenic acid species RSOH. The latter then intervenes in the $H_2S$-independent generation of a sulfheme via the transient formation of an unusual Fe(II) sulfonium species with atypical tolerance towards oxidation by $O_2$ or $H_2O_2$. Notably, while the formation of a sulfheme species from compound II and $H_2S$ is envisioned to take place through the involvement of a sulfhydryl (S• −) radical[46], our observations reveal a new mechanistic avenue with the involvement of the elusive oxadisulfane (HSOH) species during this process (Supplementary Fig. 9). Our results also indicate that HCys-induced sulfcatalase formation is pathologically relevant in a subset of cancer cells and a model of Parkinson's disease (Fig. 7). Our study therefore indicates the importance of developing therapeutic agents targeting HCys and redox-active transition metal ions to prevent the deleterious effects resulting from the combination of both.

## Methods
**Materials.** All chemicals were purchased from Sigma-Aldrich and used as-is. Bovine liver catalase was also purchased from Sigma-Aldrich. Hydrogen peroxide solution (9.79 M) TraceSELECT Ultra was purchased from Fluka. Dimedone was purchased from Acros. NaSH was purchased from Alfa Aesar in its anhydrous

form and stored in a glove box (<1 p.p.m. $O_2$ and <1 p.p.m. $H_2O$). Stock solutions of NaSH were prepared immediately before use in a buffer (50 mM phosphate buffer at pH 7.4, 1 mM DTPA) devoid of trace elements and degassed with argon.

**Enzymatic assays.** CAT activity was measured on an Uvikon 941 spectrophotometer equipped with a temperature controlled water bath ( ± 1 °C) by following the disappearance of $H_2O_2$ at 240 nm over 30–120 s at 25 °C. Catalase activity was calculated based on the rate of decomposition of hydrogen peroxide, which is proportional to the reduction of the absorbance at 240 nm. Experiments were carried out in a 3 ml quartz cuvette containing 11.7 mM $H_2O_2$ in 50 mM KPi, pH 7.4 ± 1 mM DTPA and the reaction was initiated by adding 0.3–0.6 pmoles of CAT. The effect of biological thiols on the activity of CAT under physiological conditions was determined from a solution of CAT (30–60 nM) preincubated at 25 °C for at least 120 min as a function thiol concentration (0–40 mM for HCys and Cys, 0–100 mM for GSH) in 50 mM KPi, pH 7.4, 1 mM DTPA. Control experiments (biological thiols alone) were also performed in parallel. In each case, 10 μl of the reaction mixture (biological thiols ± CAT) was added to the cuvette containing the $H_2O_2$ solution to initiate the reaction. Similarly, the effect of biological thiols on the activity of CAT under pathological conditions was determined from a solution of CAT (30–60 nM) containing a 2.124 ± 0.386 molar excess of iron that was preincubated with various concentrations of thiols (0–1 mM) for at least 120 min at 25 °C in 50 mM KP at pH 7.4. The relative half inhibitory concentration ($IC_{50}$) values were obtained by plotting the relative activity of CAT (A, in per cent) as a function of the concentration of thiol-compounds (RSH) and by fitting the data with the following four parameter logistic equation: $A = A_{min} + (A_{max} - A_{min})/(1 + ([RSH]/IC_{50})^n)$, where $A_{max}$ is the maximal activity of CAT (constrained at 100%), $A_{min}$ is the minimum activity achieved at saturating concentration of RSH and n is the Hillslope that characterizes the slope of the curve at its midpoint. The inactivation rate ($k_{inact}$, min$^{-1}$) was extracted from the time course of the inactivation of CAT bioactivity using a fixed concentration of HCys. Accordingly, a solution of CAT (3.5–5 μM) was incubated with 2 mM L-HCys in 50 mM KPi, pH 7.4, 1 mM DTPA. Aliquots (10 μl) were taken at intervals (0–480 min), diluted 1:50 in 50 mM KPi, pH 7.4, 1 mM DTPA and assayed for CAT activity as described above. Similar experiments were performed in the presence of DMPO, 5,5-dimethyl-1,3-cyclohexanedione (dimedone) or SOD at a final concentration of 100 mM, 4 mM or 40 U, respectively. $k_{inact}$ values were obtained by plotting the relative activity of CAT (A, in per cent) as a function of the incubation time and the data were fitted with an exponential decay function: $A = A_0 + \Delta A \times e^{-k_{inact} \times t}$, with $A_0$ and $\Delta A$ ($A_0 + \Delta A = 100\%$) the residual activity and the percentage of CAT inhibition when $t \to \infty$, respectively. The effect of NaSH on the consumption of $H_2O_2$ in cell lysates was determined from a solution of cell lysates (0.5–3 mg ml$^{-1}$) incubated on ice with various concentrations of NaSH (0–200 μM). The disappearance of $H_2O_2$ was then followed at 240 nm for 2–5 min at 25 °C after the addition of 20–25 μl of the reaction mixture in a 3 ml cuvette containing 11.7 mM $H_2O_2$ in 50 mM KPi, pH 7.4. The relative half inhibitory concentration ($IC_{50}$) values for NaSH were obtained by plotting the relative rate of disappearance of $H_2O_2$ ($\Delta A_{240 nm}$/min) as a function of the concentration of NaSH and by fitting the data with an hyperbolic decay equation. All the experiments were performed at least in triplicate.

**Ultraviolet–visible spectrophotometric studies.** Ultraviolet–visible spectra were recorded on a Cary 300 Scan or an Uvikon 941 spectrophotometer equipped with a temperature equilibrating water bath ( ± 1 °C). Spectral changes over time were recorded when CAT was incubated with L-HCys (0.2 or 2 mM), L-Cys (0.2, 2 or 9 mM) or GSH (1 or 30 mM) at 25 °C in a 1 ml quartz cuvette containing a solution of CAT (2.7–5.9 μM) in 50 mM KPi, pH 7.4 ± 1 mM DTPA. Due to half-site reactivity of biological thiols with CAT, the data were recorded versus a blank containing half the concentrations of CAT used in the experiments, that is, a solution of 1.35–2.95 μM CAT in 50 mM KPi at pH 7.4 ± 1 mM DTPA. Reactions

were initiated by addition of the thiol-containing compounds. Comparative kinetic studies were performed by incubating CAT (1.7–3.0 $\mu$M in the experiment, 1.35–1.5 $\mu$M in the blank) with 2 mM L-HCys in 50 mM KPi, pH 7.4, 1 mM DTPA with or without (0.1 M), 5,5-dimethyl-1,3-cyclohexanedione (dimedone, 4 mM), BCN (0.1 mM) or SOD, 40 U). All the experiments were performed at least in duplicate.

**HPLC–mass spectrometry analysis.** HPLC–MS or –HRMS spectra of the prosthetic heme-iron group were recorded on a Thermo-Finnigan Surveyor or a Thermos Fisher Accela equipped with an XTerra MS C18 3.5 $\mu$m (2.1 $\times$ 50 mm) and a pre-guard column coupled to an ESI LCQ Advantage or an Exactive orbitrap spectrometer, respectively. The prosthetic heme group was extracted on ice by incubating a solution of CAT (5–8 $\mu$M) with 1.65 volume of 50 mM Gly-HCl pH 2.0 and 2.65 vol of butan-2-one kept a $-20\,^\circ$C. After vigorous mixing and a short centrifugation step (13,000 r.p.m., 4 $^\circ$C, 30 s), the upper organic phase was quickly removed and mixed with a solution of 2 M imidazole (10% v/v) to stabilize the extracted cofactor and permit analysis. The HPLC separation was performed at 0.2 ml min$^{-1}$ with a mobile phase comprised of 0.1% formic acid in $H_2O$ (A) and 0.1% formic acid in MeOH (B) using the following steps: 40% B (0–3 min), 40–70% B (3–13 min), 70–95% B (13–14 min), 95% B (14–18 min), 95–40% B (18–19 min) and 40% B (19–20 min). Under these conditions, the prosthetic heme-iron group and the sulfheme group display overlapping HPLC peaks with retention times of 13.51–13.62 and 13.70–13.77 min, respectively. For the following protocols, the samples were first subjected to a filtering step using a Microcon filter unit of 10 kDa (40 min, 4 $^\circ$C, 11,000 r.p.m.). The HPLC separations of the reaction products and of the dimedone adducts were performed by elution at 0.1 ml min$^{-1}$ using a Satisfaction RP18AB C18 3 $\mu$m (15 $\times$ 2 mm; Cluseau) column and the following steps: 10% B (0–10 min), 10–100% B (10–50 min), 100% B (50–55 min), 100–10% B (55–55.1 min), 10% B (55.1–60 min), where A = 10 mM ammonium acetate buffer, pH 4.6 and B = ACN/MeOH/$H_2O$ (7/2/1). Under these conditions, the retention times for dimedone, dimedone-HCys, homocystine, vinylglycine/2-aminobutyric acid and $\alpha$-ketobutyrate were 17.0, 4.9, 10.1, 20.0 and 6.5 min, respectively. The HPLC separation of the BCN derivatives was achieved by elution at 0.2 ml min$^{-1}$ on a Satisfaction RP18AB C18 3 $\mu$m (15 $\times$ 2 mm; Cluseau) column using the following steps: 0–20% B (0–15 min), 20% B (15–25 min), 20–60% B (25–30 min), 60% B (30–50 min), 60-0% B (50–51 min), 0% B (51–60 min), where A = 0.1% formic acid in $H_2O$ and B = ACN. Under these conditions, the retention times for BCN-RS and BCN-RS$^+$O$^-$ were 25.5 and 18.1 min, respectively. Control experiments (CAT, thiol-compounds and additives alone as well CAT and additives or thiol-compounds and additives) were performed in each instance and analysed in parallel.

**Measurement of $H_2S$ production.** The ISO-H2S-2 (world precision instruments; WPI) polarizing voltage was set at 150 mV with a free radical analyzer (Apollo 1000; WPI). The sensor was calibrated before each experiment with freshly prepared sodium sulfide stock solution (2–10 $\mu$M), using the same buffer and conditions as the experiment. Experiments were performed for 8 h with a solution of CAT (2.8 $\mu$M) incubated with 2 mM L-HCys in 50 mM KPi, pH 7.4, 1 mM DTPA or 0.2 mM L-HCys in 50 mM KPi, pH 7.4.

**Animals.** Seven-week-old male C57BL/6J mice ($n = 6$; Envigo, Gannat, France) were acclimated for 1 week with free access to standard mouse chow and tap water. Each mouse was maintained in a cage under controlled conditions of temperature (23 $^\circ$C), humidity (55 $\pm$ 10%) and light (12:12 h light–dark cycle). Colitis was induced by the addition of DSS (3.5% (wt/vol), 36,000–50,000 MW, MP Biomedicals Illkirch-Graffenstaden, France) to the drinking water for 5 days ($n = 3$). Healthy control animals ($n = 3$) received fresh tap water, only. Two days after DSS arrest (day 7), mice were killed. All aspects of the present protocol are in accordance with the guidelines of the French Committee for Animal Care and the European convention of vertebrate animals used for experimentation under European council directive, and received written agreement from the Ministry of Higher Education and Research (APAFIS#4170-2016012213414797v3). No randomization was used and no blinding was done during this study.

**Cell culture.** Cells were obtained from American Type Culture Collection. MDST8 colorectal cancer cells were obtained from Sigma Aldrich. Unless otherwise specified, cells ($n \geq 2$) were grown to confluence in a 175 cm$^2$ culture Flask in Dulbecco's modified Eagle's medium (DMEM, containing 4,500 mg l$^{-1}$ glucose, 2.0 mM L-glutamine and 110 mg l$^{-1}$ sodium pyruvate) supplemented with 10% (v/v) heat inactivated foetal bovine serum (FBS) and maintained in a humidified 5% $CO_2$ atmosphere at 37 $^\circ$C. All cells were detached with trypsin. After trypsinolysis and two successive washes with phosphate-buffered saline (PBS) containing 0.5 g l$^{-1}$ trypsin and 0.2 g l$^{-1}$ EDTA, the cells were centrifuged at 1,400 r.p.m. for 5 min and the resulting pellet was stored at $-80\,^\circ$C. HCT-15 and HCT-116 cells were grown in RPMI-1640 Medium and McCoy's 5a Medium Modified, respectively. HT-29 Glc $-/+$ cells were cultured in a glucose free medium for 36 passages and then grown in DMEM supplemented with 10% (v/v) heat inactivated FBS in a 5% $CO_2$ humidified incubator at 37 $^\circ$C. The culture medium was changed every day. Human breast (cancer) cells were grown in DMEM supplemented with 10 mM

nonessential amino acids, 2 mM L-glutamine, 1 $\mu$g ml$^{-1}$ insulin and 10% FBS. Cells were passaged no more than 10 times after being procured from the company and their genetic characteristics were tested regularly. In addition, the presence of mycoplasma was frequently checked with the MycoAlert mycoplasma detection kit (LT07–318) from Lonza (Basel, Switzerland). For experimental purposes, cells were allowed to seed overnight before all treatments.

**Crypt and surface epithelial cell isolation.** The mice treated ($n = 3$) or not ($n = 3$) with DSS as described above were killed by cervical dislocation and the colon was removed. The faeces were flushed out with Hank's balanced salt solution (HBBS) without $Ca^{2+}$ and $Mg^{2+}$ (Biosera). The proximal end of the colon was gently everted and filled with HBSS without $Ca^{2+}$ and $Mg^{2+}$. The colon was then vortexed to remove remaining debris. Thereafter, the colon was incubated for 30 min in a solution of HBSS pH 7.4, 20 mM EDTA maintained at 37 $^\circ$C in a water bath. The tissue was then transferred in 30 ml HBSS without $Ca^{2+}$ and $Mg^{2+}$ and vortexed to release the crypts as well as the epithelial cells from the surface. The mixture was then centrifuged at 200 g for 3 min, the supernatant was discarded and the colonic crypt and the colonocytes were resuspended in 10 ml DMEM. After centrifugation at 200 g for 3 min, the pellet was washed twice with DMEM then twice with PBS and was stored at $-80\,^\circ$C.

**Transfection of Hek 293T cells and treatment of BE2-M17 cells.** Hek293T cells ($n = 2$) were plated and transfected the following day with pcDNA (empty vector), Htt-N171-82Q (Htt) or Alpha-synuclein-A53T ($\alpha$-Syn) plasmids (20 $\mu$g) with the calcium phosphate method. BE2-M17 cells ($n = 2$) were plated and treated the following day with 100 nM rotenone dissolved in DMSO or DMSO alone. Hek293T and BE2-M17 cells were collected 48 h after transfection or treatment with DMSO $\pm$ rotenone, respectively.

**Western blot analysis.** To check the expression of Htt and $\alpha$-Syn in Hek293T cells 48 h after transfection, Western blot analyses were performed. After being washed once with PBS, cells were lysed on ice for 30 min in 50 mM Tris–HCl, pH 7.5, 150 mM NaCl, 1% NP40, 1 mM phenylmethylsulphonyl fluoride and 1% protease inhibitor cocktail. After centrifugation (10 min, 15,000 g, 4 $^\circ$C) the supernatants were collected and protein concentration was assessed with the BCA kit (Thermo Fisher). 20 $\mu$g of total proteins were loaded onto a 4–12% SDS-PAGE gel (NuPAGE, Thermo Fisher) after denaturing in loading buffer (5 min, 95 $^\circ$C). The proteins were then transferred on a nitrocellulose membrane (iBlot, Thermo Fisher Scientific). The membranes were first incubated 1 h at RT with agitation in 20 mM Tris–HCl, pH 7.5, 150 mM NaCl, 0.1% Tween-20 (TBS-T) supplemented with 5% nonfat dry milk (TBS-T $+$ 5%). Next, the membranes were incubated 3 h at RT with agitation with primary antibodies (mouse anti-Htt (Merck Millipore), mouse anti-$\alpha$-Syn (ThermoFisherScientific), mouse anti-Tubulin (Sigma-Aldrich)) diluted at 1/1,000 in TBS-T $+$ 5%. After 4 washes with TBS-T, membranes were incubated with HRP-conjugated secondary antibody (anti-mouse, GE Healthcare) prepared in TBS-T for 1 h at RT on agitation, and washed again 4 times with TBS-T. The detection of the proteins was performed with ECL Clarity (Bio-Rad) and the Fusion FX7 (Fisher Bioblock Scientific).

**Endogenous catalase activity in cell lysates.** The relative specific activity (activity and quantity) of catalase in the cell lysates was determined using the kit ab118184 from Abcam (Cambridge, MA). The measurements were performed in duplicate as per the manufacturer's protocol. Briefly, catalase in cell lysates (0.1–0.2 mg cell lysates per assay) was immunocaptured within the wells of the microplate. An assay buffer, which contains $H_2O_2$ that reacts with a substrate to generate a luminescent product, was added to each well. The presence of catalase in the reaction mixture reduces the production of light and the light produced in each well is thus inversely proportional to the amount of catalase activity. Then, the quantity of catalase was measured by adding to each well an anti-catalase primary detector antibody. After 1 h incubation at RT, the unbound detector antibody was washed away and an HRP-conjugated labelled secondary antibody specific for the primary detector antibody was added to the wells. After 1 h incubation, the wells were again washed and a 3,3',5,5'-Tetramethylbenzidine (TMB) substrate solution was added to the wells. A blue colour (absorbance at 600 nm) developed in proportion to the amount of catalase bound. Luminescence and absorbance measurements were performed on an EnSpire Multilabel Reader 2300 (PerkinElmer). To check the effect of NaSH on the relative specific activity of catalase, the immunocaptured catalase was incubated with various concentrations of NaSH (0–50 $\mu$M) for 5 min at RT before carefully washing twice each well. The protocol above was then followed to determine activity measurement and catalase quantification.

**Analysis.** Protein concentration was determined by the method of Bradford (BioRad) or with the BCA kit (Thermo Fisher). The iron content was determined under reducing conditions with bathophenantroline disulfonate (Alfa Aesar) after acid denaturation of the enzyme. Briefly, 65 $\mu$l of CAT (3–10 mg ml$^{-1}$) were denatured with 45 $\mu$l perchloric acid 1 M. After 1 h at RT, the samples were centrifuged for 5 min at 15,000 r.p.m. Bathophenantroline disulfonate (72 $\mu$l at

1.70 mg ml$^{-1}$), sodium ascorbate (36 µl at 38.0 mg ml$^{-1}$) and sodium acetate (27 µl of a saturated solution diluted 1:3) were successively added to 90 µl of the supernatant. After 30 min at RT and centrifugation (5 min at 15,000 r.p.m.), the absorbance (538–680 nm) was measured on a Cary 300 Scan. The iron content was calculated from a standard curve obtained with an ammonium iron(II) sulfate hexahydrate (Sigma) solution.

**Data availability**. All the relevant data are available from the authors.

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

## Acknowledgements

We thank Dr Emmanuel Brouillet (MIRCen) and Dr Pierre Laurent-Puig (Université Paris Descartes) for useful discussion on the various pathological models. We are indebted to Dr Diana Over (Université Paris Descartes) and Dr Keith Hall (Senzex Corporation, USA) for carefully reading and editing the manuscript.

## Author contributions

D.P. designed and performed experiments, analysed data and wrote the manuscript. A.H. performed mass spectrometry analysis. F.T.C. and S.S. performed all experiments on HBC cells. G.L. provided cellular models for neurodegenerative diseases. M.A., A.L. and F.B. participated in the choice of the colitis model, provided animals, isolated crypts and surface epithelial cells and cultured HT-29 Glc −/+ cells. C.P. grew all other colorectal cancer cells. E.G. and I.A. helped analyse data and write the manuscript.

## Additional information

**Competing financial interests:** The authors declare no competing financial interests.

