## [Peer Review File · Nature Communications]

Reviewer #1 (Remarks to the Author):

In this work, the authors use activity assays and mass spectrometry, mainly, to investigate the effect of several thiols on the heme prosthetic group of bovine liver catalase. It is shown that sulfheme can be produced by incubation with homocysteine. Several mechanisms for this formation are tested by the inclusion of metal chelators, radical scavengers, and superoxide dismutase. Ex cellulo studies from a human breast cancer cell line are also employed in an attempt to tie the mechanistic findings to a relevant human disease. This work attempts to expand the scope of heme modification by identifying an unprecedented route for the formation of sulfheme that does not utilize H₂S but HCys instead. The impact of the finding is potentially high because of the possible link of the modification to different pathologies. I think the link to various diseases mentioned in the manuscript needs more convincing data. Overall, the authors have found something interesting, however, the presentation is very difficult to follow. I think the authors should take more space to fully explain their data. Also, I would recommend an English editing service. In its present version, the manuscript needs an overhaul before it can be recommended for publication.

Specific Concerns:

1. It is a little strange that this work was done with catalase purchased from sigma without further purification.
2. The authors have shown that they are capable of recording EPR spectra, and that the reaction course is in the minutes to hours time scale, Figure 1. It is expected then that reaction is quenched at varying times and its EPR spectrum measured to show the changing redox state of the heme (the first and second intermediates are expected to be EPR silent and the final state should have a distinct signal). Even though they gave several reasons for not catching HCys• via EPR, it will be strengthened if they were utilizing EPR to show the change of not only radicals but also heme irons through the whole reaction course. They indicated this in experimental section, but no data was shown elsewhere.
3. The data in Figures 1 and 2 seem to be inconsistent. In figure 1, it looks as though a very large fraction of the heme reacts with HCys, however, the mass spec results in Figure 2 seems to show a relatively low conversion efficiency.
4. The reaction conditions for the various experiments are not well defined for the reader, enzyme concentration, presence / concentration of peroxide?
5. Fe(V) while being chemically equivalent to Fe(IV) + por* (Compound I) they are not the same species and should be treated as such. In equation [1-2], expression for compound I should be CAT-Fe(IV)=O(.+) rather than CAT-Fe(V)=O. As is it is compound II, a one-electron reduced form of the compound I.
6. There are 2 carbon ethylene linkage in a Heme B group. This work left an open question why only one carbon-carbon double bond was modified, and which one was modified. Was the target determined randomly or dependent on the chemical environment?
7. Figure S1 shows that cysteine has the largest effect on catalase activity. Figure S9 shows that >100 mins are needed for maximum effect. Was the enzyme in S1 incubated for >100 mins before the assays were performed? If not, is the inhibitory effect in S1 due to something other than sulfheme formation?
8. It is not defined what the concentration of the 'trace elements' is. Are they in a physiological range?

9. Page 8, Fe(II) species shows alpha-bands at 591, 636 and 655 nm but, in Figure 1B, why 658 instead of 655?
10. S-oxidation and S-oxygenation were both used throughout the manuscript, which is confusing.
11. Page 14, Figure S10D and Figure S10C should be corrected to S11D and S11C, respectively.
12. No potential explanation is given for why the MDA-MB-231 cell lysate is insensitive to NaSH. With no 'healthy' cell line for comparison, this data is confusing and potentially contradictory to the central hypothesis of the manuscript.
13. As is, too much relevant experimental detail is not included and there is far too heavy reliance on data in the SI.

Reviewer #2 (Remarks to the Author):

A. Summary of the key results

The key result is that CAT is able to oxidize thiols to thyl radical that mediate formation of the CAT-Fe(IV)=O species (compound II), which inactivates the enzyme. This S-oxygenation reaction highlights the role of Compound II in thiol oxidation biochemistry.

B. Originality and interest: if not novel, please give references

The authors use a variety of spectroscopic and analytical methods to support their investigations. Interest in catalase activity, thiol modifications, and postsynthetic heme modifications are of interest to researchers in chemistry, biology, and biochemistry. Overall, the manuscript is original, but also in a very specific niche associated with redox biology and thiol oxidation.

C. Data & methodology: validity of approach, quality of data, quality of presentation

The approach is valid.

-Showing the data (in the SI) for the important negative controls in the "Sulfeme formation occurs without H₂S intervention" section would be appropriate.

Comments on quality of presentation:

-The manuscript needs to be edited carefully. There are many typographical and grammatical errors in the manuscript. The authors also need to carefully check that all references to Figures match the data in the figures. There were a few cases in the manuscript where this was not the case.

-Figure S2: The g-values should be labeled

-Figure S9: "Dimedone" is misspelled in the figure.

-Figure S11: In (A) - the 25 micromolar and 0.3 mM should indicate that these sulfide quantities are added to the Fe compound (i.e. they are not just UV-vis spectra of the sulfide salts alone. Also, in (B), the figure caption needs clarification. This spectrum appears to be a difference spectrum - referring to this as a differential spectrum may suggest to some readers that this is a first derivative plot of the data. Also, it is unclear what the difference spectra are being referenced/subtracted from.

-Figure S12: There is a break in the data. I am assuming that the lower energy absorbances are multiplied by some factor, but this multiplication factor is not indicated in the figure or caption.

D. Appropriate use of statistics and treatment of uncertainties

N/A

E. Conclusions: robustness, validity, reliability

-Does DTPA inhibit CAT directly? (i.e. rather than just removing trace metals does DTPA interact with CAT directly). What if the buffer is pre-treated with a metal-removing agent, such as Chelex,

to remove the metals prior to CAT addition- Is the same inhibition observed?

-The authors comment on the impacts of pKa and steric hindrance from testing only Cys, Hcy, and GSH. These claims should be further supported (i.e use of other thiols to test this hypothesis) or revised.

F. Suggested improvements: experiments, data for possible revision.

In the introduction, the statement "The presence of trace elements, used to mimic pathological conditions, exacerbates the inactivation of CAT bioactivity" should either be clarified significantly or removed. Which elements? Also, "exacerbates the inactivation" is unclear.

The phrase "extra temporally" is used throughout the manuscript, but it is unclear to this reviewer what this phrase means (other than outside of time, which doesn't make sense in the context of the manuscript or in reference to cortical surgery, which also doesn't make sense in this context...)

G. References: appropriate credit to previous work?

Appropriate

H. Clarity and context: lucidity of abstract/summary, appropriateness of abstract, introduction and conclusions

The conclusions describes one of the key findings to be "an intervention of compound II into an S-oxygenation reaction" but this is not reflected in the abstract.

In the opinion of this reviewer, the comments in the conclusion that HCys-induced sulfheme formation is physiologically-relevant in cancers, Alzheimers, PD, and Huntington's Disease and that sulfheme formation by other, more standard mechanisms, is doubtful in these cases is an overstatement of the conclusions supported by the data presented in the paper.

Reviewer #3 (Remarks to the Author):

In this report a detailed study is presented on the reaction of catalase with homocysteine, which has been proposed to be important in the activity of neurodegenerative diseases. The authors find a novel pathway, whereby homocysteine reacts with the active species of catalase (possibly compound II) and attacks the heme and convert it to sulfheme. This is an original idea and observation that may explain the pathological evidence. The work is of interest and Nature Commun is the appropriate journal for this work. The experiments have been done using appropriate methods and techniques and publication is recommended.

I don't think an iron(V) is formed in catalases, I would expect an iron(IV)heme cation radical instead (Eqs 1 & 2). See Shaik et al Chem. Rev. 2005.

It would also be good to cite the work of Kumar et al on substrate sulfoxidation by heme proteins (Chem. Eur. J. 2011, pp 6196)

Abstract: "In these later", rewrite.

Page 9: typo in sulfheme.

Reviewer #1 (Remarks to the Author):

In this work, the authors use activity assays and mass spectrometry, mainly, to investigate the effect of several thiols on the heme prosthetic group of bovine liver catalase. It is shown that sulfheme can be produced by incubation with homocysteine. Several mechanisms for this formation are tested by the inclusion of metal chelators, radical scavengers, and superoxide dismutase. Ex cellulo studies from a human breast cancer cell line are also employed in an attempt to tie the mechanistic findings to a relevant human disease. This work attempts to expand the scope of heme modification by identifying an unprecedented route for the formation of sulfheme that does not utilize H₂S but HCys instead. The impact of the finding is potentially high because of the possible link of the modification to different pathologies. I think the link to various diseases mentioned in the manuscript needs more convincing data. Overall, the authors have found something interesting, however, the presentation is very difficult to follow. I think the authors should take more space to fully explain their data. Also, I would recommend an English editing service. In its present version, the manuscript needs an overhaul before it can be recommended for publication.

We would like to thank this referee for his useful comments. We do believe that we answered all the referee's concerns (see below). As such, the manuscript has been significantly improved. We now took more space to explain some points. To strengthen the link between HCys-induced sulfcatalase formation and the various diseases mentioned in the manuscript, we performed *ex cellulo* experiments on various human breast cancer cells, numerous colorectal cancer cells, diverse cellular models of neurodegenerative disorders (Hek 293T cells transfected with Htt-N171-82Q or Alpha-synuclein-A53T, and M17 human neuroblastoma cells treated with rotenone) and a rodent model of Dextran Sulfate Sodium(DSS)-induced colitis. We explored *ex cellulo* the relative specific activity of CAT (SpCAT), monitored the effect of NaSH on the rate of H₂O₂ consumption in cell lysates and studied the effect of NaSH on SpCAT to allow discriminating between various inactive states of the enzyme. In addition, the manuscript has now been carefully edited.

Specific Concerns:

1. It is a little strange that this work was done with catalase purchased from sigma without further purification.

We were unaware of the impurities contained in the various catalase batches purchased from Sigma, even in the most active aqueous suspension form. We realized this once our experiments with DTPA were achieved, by performing activity tests in the absence of DTPA, EPR measurements as well as proteomic studies. However, we are confident about our results that clearly show a direct interaction of thiols with the heme-iron of CAT and demonstrate that HCys is the sole biological thiol to induce sulfcatalase formation. In addition, we did perform *ex cellulo* studies on various cellular models of diseases that also suggest the likelihood of this scenario in some of them.

2. The authors have shown that they are capable of recording EPR spectra, and that the reaction course is in the minutes to hours time scale, Figure 1. It is expected then that reaction is quenched at varying times and its EPR spectrum measured to show the changing redox state of the heme (the first and second intermediates are expected to be EPR silent and the final state should have a distinct signal). Even though they gave several reasons for not catching HCys via EPR, it will be strengthened if they were utilizing EPR to show the change of not only radicals but also heme irons through the whole reaction course. They indicated this in experimental section, but no data was shown elsewhere.

We now added the EPR experiments performed at 10 K. Unfortunately, we were unable to characterize the sulfheme species by EPR spectroscopy due to the half-site reactivity of CAT with HCys. Hence, the High Spin EPR signal resulting from the reactivity of CAT with HCys clearly differs from the one observed with native CAT-Fe(III) alone but appears to be a mixture of several High Spin species (Supplementary Fig. 6).

3. The data in Figures 1 and 2 seem to be inconsistent. In figure 1, it looks as though a very large fraction of the heme reacts with HCys, however, the mass spec results in Figure 2 seems to show a relatively low conversion efficiency.

In our hand, the heme-iron prosthetic group extracted with butan-2-one under acidic conditions is unstable. We did mention it in the original manuscript (Methods). As such, we also mentioned that we stabilized the heme-iron prosthetic group with imidazole immediately after extraction. This crucial piece of information has now been added in the main text (Supplementary Fig. 5). The apparent “relatively low conversion efficiency” shown in Figure 2 may thus result from the instability of the heme-iron after extraction. Also, in our case, the mass spectrometry results are not quantitative. The signal intensity observed by HRMS or MS is dependent on the ionization of the product.

4. The reaction conditions for the various experiments are not well defined for the reader, enzyme concentration, presence / concentration of peroxide?

This point has now been clarified. We added all the concentrations of the enzyme throughout the paper. The concentration of peroxide used in the activity tests has been changed from percentage to molarity. We would like to add that we do not add any peroxide in our experiments, apart from the activity tests. Hence, the hydrogen peroxide necessary for sulfheme formation comes from thiyl radicals ($RS^{\bullet} + RS^{-} \rightarrow [RSSR]^{\bullet}$; $[RSSR]^{\bullet} + O_2 \rightarrow RSSR + O_2^{\bullet}$; dismutation of O_2^{\bullet} to H_2O_2) either generated by interaction of CAT-Fe(III) with thiols or by thiols that enter a futile redox cycle in the presence of redox active transition metal ions.

5. Fe(V) while being chemically equivalent to Fe(IV) + por (Compound I) they are not the same species and should be treated as such. In equation [1-2], expression for compound I should be CAT-Fe(IV)=O(.+) rather than CAT-Fe(V)=O. As is it is compound II, a one-electron reduced form of the compound I.*

This point has been corrected.

6. There are 2 carbon ethylene linkage in a Heme B group. This work left an open question why only one carbon-carbon double bond was modified, and which one was modified. Was the target determined randomly or dependent on the chemical environment?

This is an interesting question as several isoforms could exist, as observed in sulfmyoglobin (episulfide, ring opened episulfide and thiochlorin). However, to answer to this question, we will need to carry out NMR experiments. This is unfortunately impossible due to the instability of the extracted heme-iron prosthetic group (see point #3).

7. Figure S1 shows that cysteine has the largest effect on catalase activity. Figure S9 shows that >100 mins are needed for maximum effect. Was the enzyme in S1 incubated for >100 mins before the assays were performed? If not, is the inhibitory effect in S1 due to something other than sulfheme formation?

Yes, the enzyme in S1 was incubated for more than 100 mins before the assays were performed, as described in Methods. The reaction conditions for the various experiments are now better defined in

Methods. Furthermore, we made clear in the main text that enzyme inhibition is initially due to compound II formation that represents a temporarily inactive state of the enzyme, in contrast to sulfheme formation that represents an irreversibly inactive state of the enzyme.

8. *It is not defined what the concentration of the 'trace elements' is. Are they in a physiological range?*

We performed analyses of the iron content and replaced “trace elements” by “iron”. Iron concentrations appear to be in the pathological range (55-145 nM).

9. *Page 8, Fe(II) species shows alpha-bands at 591, 636 and 655 nm but, in Figure 1B, why 658 instead of 655?*

The alpha-band around 655-658 nm is sometimes more a shoulder than a band (see for instance Supplementary Fig. 2a). We now used 658 nm throughout the manuscript, based on Fig 1B where the α -band is well-defined.

10. *S-oxidation and S-oxygenation were both used throughout the manuscript, which is confusing.*

We now used S-oxygenation instead of S-oxidation as the former is more specific than the latter.

11. *Page 14, Figure S10D and Figure S10C should be corrected to S11D and S11C, respectively.*

Supplementary Figures have been changed but the edits have been performed.

12. *No potential explanation is given for why the MDA-MB-231 cell lysate is insensitive to NaSH. With no 'healthy' cell line for comparison, this data is confusing and potentially contradictory to the central hypothesis of the manuscript.*

We now performed *ex cellulo* experiments on various cellular models of diseases with appropriate control cell lines (Fig. 5). We monitored the effect of NaSH on the rate of H₂O₂ consumption in cell lysates and we observed an inhibition of the rate of H₂O₂ disappearance by NaSH in all cell lysates. These results suggest that H₂S inhibits ubiquitously cellular antioxidant enzymes that rapidly consume H₂O₂ most likely *via* the formation of cysteine persulfides that affect the activity of target proteins, as recently reported for peroxiredoxin-1 and peroxiredoxin-6 (reference 71). In addition, we also monitored the influence of NaSH on the relative specific activity of CAT (SpCAT), which allowed discriminating between various inactive states of the enzyme. Hence, CAT is solely inhibited *via* compound II formation in a cancer cell line (MDST8 cells) while it is inhibited *via* sulfheme formation in other cancer cell lines and a model of Parkinson's disease.

13. *As is, too much relevant experimental detail is not included and there is far too heavy reliance on data in the SI.*

We do fully agree with this comment. We improved the experimental section, reorganized all our data and moved the most relevant ones in the main text. We now have five figures instead of four. One with the activity tests (Fig. 1), one with all the relevant data concerning sulfheme formation induced by HCys (Fig. 2), one with the comparative kinetic studies in the presence of various additives as well as the HRMS characterization of trapped reactive species (Fig. 3), one with the comparative reactivity of various heme-iron species towards oxidants (Fig. 4) and one with *ex cellulo* studies (Fig. 5).

Reviewer #2 (Remarks to the Author):

We would like to thank this referee for his helpful comments. We took into consideration the referee's comments to strengthen and correct the manuscript.

A. Summary of the key results

The key result is that CAT is able to oxidize thiols to thyl radical that mediate formation of the CAT-Fe(IV)=O species (compound II), which inactivates the enzyme. This S-oxygenation reaction highlights the role of Compound II in thiol oxidation biochemistry.

B. Originality and interest: if not novel, please give references

The authors use a variety of spectroscopic and analytical methods to support their investigations. Interest in catalase activity, thiol modifications, and postsynthetic heme modifications are of interest to researchers in chemistry, biology, and biochemistry. Overall, the manuscript is original, but also in a very specific niche associated with redox biology and thiol oxidation.

C. Data & methodology: validity of approach, quality of data, quality of presentation

The approach is valid.

-Showing the data (in the SI) for the important negative controls in the "Sulfeme formation occurs without H₂S intervention" section would be appropriate.

We do agree. This has been done (Supplementary Fig. 8)

Comments on quality of presentation:

-The manuscript needs to be edited carefully. There are many typographical and grammatical errors in the manuscript. The authors also need to carefully check that all references to Figures match the data in the figures. There were a few cases in the manuscript where this was not the case.

The manuscript has now been carefully edited.

-Figure S2: The g-values should be labeled

All the g-values have been added to the Figure showing our EPR experiments (Supplementary Fig. 6).

-Figure S9: "Dimedone" is misspelled in the figure.

Thank you. This has been corrected.

-Figure S11: In (A) - the 25 micromolar and 0.3 mM should indicate that these sulfide quantities are added to the Fe compound (i.e. they are not just UV-vis spectra of the sulfide salts alone. Also, in (B), the figure caption needs clarification. This spectrum appears to be a difference spectrum - referring to this as a differential spectrum may suggest to some readers that this is a first derivative plot of the data. Also, it is unclear what the difference spectra are being referenced/subtracted from.

We changed this figure (now Fig. 4 in the main text). Instead of Fig S11B, we now added all the observed values for the alpha-bands and the Soret band.

-Figure S12: There is a break in the data. I am assuming that the lower energy absorbances are multiplied by some factor, but this multiplication factor is not indicated in the figure or caption.

Thank you. This was an omission.

D. Appropriate use of statistics and treatment of uncertainties

N/A

E. Conclusions: robustness, validity, reliability

-Does DTPA inhibit CAT directly? (i.e. rather than just removing trace metals does DTPA interact with CAT directly). What if the buffer is pre-treated with a metal-removing agent, such as Chelex, to remove the metals prior to CAT addition- Is the same inhibition observed?

We carried out the appropriate control experiment that shows that DTPA (0-10 mM) does not interfere with catalase activity (Supplementary Fig. 1).

-The authors comment on the impacts of pKa and steric hindrance from testing only Cys, Hcy, and GSH. These claims should be further supported (i.e use of other thiols to test this hypothesis) or revised.

We did suppress these claims as we only wanted to include Cys, HCys and GSH in the manuscript.

F. Suggested improvements: experiments, data for possible revision.

In the introduction, the statement "The presence of trace elements, used to mimic pathological conditions, exacerbates the inactivation of CAT bioactivity" should either be clarified significantly or removed. Which elements? Also, "exacerbates the inactivation" is unclear.

We now performed analyses and clarified this point. In our case, "trace elements" is "iron".

The phrase "extra temporally" is used throughout the manuscript, but it is unclear to this reviewer what this phrase means (other than outside of time, which doesn't make sense in the context of the manuscript or in reference to cortical surgery, which also doesn't make sense in this context...)

Indeed. We changed it and wrote "immediately prior to use".

G. References: appropriate credit to previous work?

Appropriate

H. Clarity and context: lucidity of abstract/summary, appropriateness of abstract, introduction and conclusions

The conclusions describes one of the key findings to be "an intervention of compound II into an S-oxygenation reaction" but this is not reflected in the abstract.

The abstract has been modified. However, the abstract from the original manuscript reflected this point: "Prosthetic heme modification results from the unprecedented S-oxidation of homocysteine by the ferryl center of compound II".

In the opinion of this reviewer, the comments in the conclusion that HCys-induced sulfcatalyse formation is physiologically-relevant in cancers, Alzheimers, PD, and Huntington's Disease and that sulfheme formation by other, more standard mechanisms, is doubtful in these cases is an overstatement of the conclusions supported by the data presented in the paper.

We do fully agree with this comment. As a result, we attempted to strengthen the link between HCys-induced sulfcatalase formation and various diseases by performing *ex cellulo* experiments on various human breast cancer cells, numerous colorectal cancer cells, diverse cellular models of neurodegenerative disorders (Hek 293T cells transfected with Htt-N171-82Q or Alpha-synuclein-A53T, and M17 human neuroblastoma cells treated with rotenone) and a rodent model of Dextran Sulfate Sodium(DSS)-induced colitis. We explored *ex cellulo* the relative specific activity of CAT (SpCAT), monitored the effect of NaSH on the rate of H₂O₂ consumption in cell lysates and studied the effect of NaSH on SpCAT. Our results now suggest the likelihood of HCys-induced sulfheme formation in some models of diseases and even point out the existence of a temporarily inactivate state of the enzyme (compound II) in one cancer cell line.

Reviewer #3 (Remarks to the Author):

We thank this referee for his comments. We took into consideration the referee's comments to correct the manuscript.

In this report a detailed study is presented on the reaction of catalase with homocysteine, which has been proposed to be important in the activity of neurodegenerative diseases. The authors find a novel pathway, whereby homocysteine reacts with the active species of catalase (possibly compound II) and attacks the heme and convert it to sulfheme. This is an original idea and observation that may explain the pathological evidence. The work is of interest and Nature Commun is the appropriate journal for this work. The experiments have been done using appropriate methods and techniques and publication is recommended.

I don't think an iron(V) is formed in catalases, I would expect an iron(IV)heme cation radical instead (Eqs 1 & 2). See Shaik et al Chem. Rev. 2005.

Thank you. This has been corrected.

It would also be good to cite the work of Kumar et al on substrate sulfoxidation by heme proteins (Chem. Eur. J. 2011, pp 6196)

This interesting paper is a DFT study on the sulfoxidation reactions mediated by models of P450 enzymes. However, this paper is centered around the capacity of compound I to perform such sulfoxidation reactions. In our paper, we are interested by substrate sulfoxidation mediated by heme-iron(IV)-oxo and not iron(IV)-oxo porphyrin cation radicals. As such, even if of great interest, we did not cite this paper.

Abstract: "In these later", rewrite.

This has been rewritten and changed by "in the latter".

Page 9: typo in sulfheme.

Thank you. This has been corrected.

Reviewer #1 (Remarks to the Author):

This revision represents a drastic improvement over the previous version. I feel that the authors have adequately addressed reviewers' concerns by rearranging their data presentation, clarifying the text, and the addition of significant ex cellulo studies. This work is recommended for publication after correction of two minor details below.

(1) There is no EPR data in the main text, so the methods section for EPR should be moved to the SI.

(2) Scheme 1, Pathway B, species 5, it is not appropriate to refer to Compound I as an Fe(V)=O. It should be an Fe(IV)=O(.+).

REVIEWERS' COMMENTS:

Reviewer #1 (Remarks to the Author):

This revision represents a drastic improvement over the previous version. I feel that the authors have adequately addressed reviewers' concerns by rearranging their data presentation, clarifying the text, and the addition of significant ex cellulose studies. This work is recommended for publication after correction of two minor details below.

(1) There is no EPR data in the main text, so the methods section for EPR should be moved to the SI.

We now did move the method section for EPR into Supplementary methods.

(2) Scheme 1, Pathway B, species 5, it is not appropriate to refer to Compound I as an Fe(V)=O. It should be an Fe(IV)=O(.+).

We do thank the referee for this comment. We did forget to modify this mistake in the scheme. This has now been done.